# The Antiproliferative Effects of Flavonoid MAO Inhibitors on Prostate Cancer Cells

**DOI:** 10.3390/molecules25092257

**Published:** 2020-05-11

**Authors:** Najla O. Zarmouh, Samia S. Messeha, Nelly Mateeva, Madhavi Gangapuram, Kacy Flowers, Suresh V. K. Eyunni, Wang Zhang, Kinfe K. Redda, Karam F. A. Soliman

**Affiliations:** 1College of Pharmacy & Pharmaceutical Sciences, College of Medical Technology-Misurata, Libyan National Board for Technical & Vocational Education, Misrata, Libya; najlazar@yahoo.com; 2College of Pharmacy & Pharmaceutical Sciences, Florida A & M University, Tallahassee, FL 32307, USA; samia.messeha@famu.edu (S.S.M.); madhavi.gangapuram@famu.edu (M.G.); kacy1.flowers@famu.edu (K.F.); suresh.eyunni@famu.edu (S.V.K.E.); wang.zhang@famu.edu (W.Z.); 3College of Science and Technology, Florida A & M University, Tallahassee, FL 32307, USA; nelly.mateeva@famu.edu

**Keywords:** flavonoid derivatives, MAO-A and MAO-B, LNCaP, DU145, prostate cancer, depression, antiproliferation

## Abstract

Prostate cancer (PCa) patients commonly experience clinical depression. Recent reports indicated that monoamine oxidase-A (MAO-A) levels elevate in PCa, and antidepressant MAO-Is show anti-PCa properties. In this work, we aimed to find potential drugs for PCa patients suffering from depression by establishing novel anti-PCa reversible monoamine oxidase-A inhibitors (MAO-AIs/RIMA); with an endeavor to understand their mechanism of action. In this investigation, twenty synthesized flavonoid derivatives, defined as KKR compounds were screened for their inhibitory potentials against human MAO-A and MAO-B isozymes. Meanwhile, the cytotoxic and antiproliferative effects were determined in three human PCa cell lines. MAO-A-kinetics, molecular docking, SAR, cell morphology, and cell migration were investigated for the most potent compounds. The screened KKRs inhibited MAO-A more potently than MAO-B, and non-toxically inhibited LNCaP cell proliferation more than the DU145 and PC3 cell lines, respectively. The results showed that the three top MAO-AI KKRs compounds (KKR11, KKR20, and KKR7 (IC_50_s 0.02–16 μM) overlapped with the top six antiproliferative KKRs against LNCaP (IC_50_s ~9.4 μM). While KKR21 (MAO-AI) and KKR2A (MAO-I) were ineffective against the PCa cells. Furthermore, KKR21 and KKR11 inhibited MAO-A competitively (K_i_s ≤ 7.4 nM). Molecular docking of the two compounds predicted shared hydrophobic and distinctive hydrophilic interactions—between the KKR molecule and MAO-A amino acid residues—to be responsible for their reversibility. The combined results and SAR observations indicated that the presence of specific active groups—such as chlorine and hydroxyl groups—are essential in certain MAO-AIs with anti-PCa effects. Additionally, MAO-A inhibition was found to be associated more with anti-PCa property than MAO-B. Distinctively, KKR11 [(*E*)-3-(3,4-dichlorophenyl)-1-(2-hydroxy-4,6-dimethoxyphenyl)prop-2-en-1-one] exhibited anti-metastatic effects on the DU145 cell line. The chlorine substitution groups might play vital roles in the KKR11 multiple actions. The obtained results indicated that the flavonoid derivative KKR11 could present a novel candidate for PCa patients with depression, through safe non-selective potent inhibition of MAOs.

## 1. Introduction

Prostate cancer (PCa) is the second most common malignancy among males. The high incidence, economic burden, and fatality rate makes PCa a global health issue [1,2,3]. PCa patients commonly experience depression and sometimes major depressive disorder (MDD). Recent estimates showed that 60% men with PCa experience depression, of which up to 40% become clinically significant [2]. Emerging meta-analysis research highlighted the prevalence of depression in PCa patients and urged further investigation [1]. Although the actual causes of depression among PCa patients are unclear, PCa sufferers experience exhausting symptoms and mortality concerns, which is manifested by an increased suicide rate [2,3], low quality of life, poor adherence to treatment, and, consequently, the poorer prognosis of PCa [4].

Currently, the most common treatment for metastatic PCa is androgen deprivation therapy (ADT), such as the anti-androgen flutamide (FLUT) and surgery, which might accompany radiation. Unfortunately, the benefit of these treatments is temporary, as many patients experience disturbing side effects from androgen deficiency [5,6]. Moreover, current ADT is the main suspect of increasing PCa patient risk of depression [7,8] and suicide rates [2]. Indeed, depression among patients receiving ADT reached three-fold higher than others who did not undergo ADT [9]. Additionally, with the increase of the PCa progression stage and aggressiveness, these events might lead to the failure of current chemotherapies. On the contrary, the use of ADT was found to later induce tumor and worsen tumor progression as the treatments promoted resistance to chemotherapy [10] and enhanced castrate-resistant prostate cancer (CRPC) [11]. The relationship of ADT to the resistant PCa heterogenic subtypes, the development of CRPC complications [12] and depression are increasingly recognized. The latest drug discoveries are holding promises to fight PCa with more drugs targeting the signaling of androgen receptors (AR), immunotherapy, and taxane-based chemotherapy [12]. However, benefits are yet clinically limited by increasing cancer resistance that allows for high mortality and morbidity rates. These limitations raised calls for the demand for more research on more efficient and less controversial possible treatments [13]. Additionally, due to these ADT tolerance limitations, better regimens, and multifunctional treatments against PCa becomes a necessity.

The high activity of monoamine oxidase (MAO) isozymes is associated with many neurodegenerative disorders, as well as several cancer types, including PCa [14]. Interestingly, PCa aggressiveness was recently associated with both depression development and the high activity of the isozyme MAO-A [15]. The high MAO-A levels are usually associated with depressive symptoms through the reduction of monoamine neurotransmitters and initiating neuronal death in the brain [16]. On the other hand, emerging research indicated that the overexpressed MAO-A was among the top changed genes related to tumorigenesis in high-grade, poorly undifferentiated PCa [17,18]. High levels of MAO-A were positively correlated with the metastasis-related receptor CD44 and the prostate-specific antigen (PSA), an indicator of early PCa [17]. It was reported that MAO-A overexpression promoted tumor growth, induced epithelial-to-mesenchymal transition, and enhanced metastasis in mice [19]. Indications of PCa defensive mechanisms by increasing cellular MAO-A in patients on chemotherapeutics were reported, as high enzyme levels increased patients’ resistance to chemotherapy drugs such as docetaxel [20].

The primary isozymes of MAO, MAO-A, and MAO-B, are mitochondrial-bound flavin-containing amine oxidoreductases that normally lower the cytoplasmic monoamine levels to maintain homeostasis [14]. As it dominates specific cells, MAO-A catalyzes excessive neurotransmitters in the CNS, and dietary amines in the intestine to prevent hypertension caused by tyramine [21]. Additionally, it is expressed by the human prostate epithelium. In the prostate gland, MAO-A is exclusively expressed in the basal and progenitor cells, preventing their differentiation into secretory epithelial cells, as these secretory cells are almost absent from MAO-A [18]. Hence, the distribution and function of MAO-A in these vital tissues highlight its role in triggering PCa and its relationship with depression.

It is well-established that the overly active MAO-A contributes to aggravating oxidative stress in many organs. The abnormal elevation of MAO-A activity consequently elevates its toxic byproducts hydrogen peroxide (H_2_O_2_) and aldehydes [22]. The overwhelming byproducts contribute to cellular oxidative stress that leads to mitochondrial toxicity, DNA, and severe lipid membrane damage—and thus—neuronal cell death and contribution to PCa [19,23]. Certainly, active MAO-A aggravates the reactive oxygen species and activates nuclear translocation and expression of hypoxia-inducible factor 1α (HIF1α) that leads to aggressive cellular phenotypes [20].Therefore, it is unsurprising that oxidative stress is found to contribute to depression and PCa in aging men [24,25].

It is becoming clear that MAO-A overexpression is associated with prostate cellular dedifferentiation, tumorigenesis, aggressiveness, metastasis, as well as depressive symptoms, which dictate its inhibition, for potential PCa treatment and depression management. Treating with MAO inhibitors (MAO-Is), particularly the reversible inhibitors of MAO-A (RIMA), is currently a well-established strategy for depression in the elderly. It certainly has been proven more effective than other antidepressant mechanisms [26,27]. Additionally, their reversibility of inhibition reduces the side effects of the unmetabolized monoamine substrates. However, their effects against cancer are discrete [27] with more focus on the non-clinical irreversible MAO-AI clorgyline [18]. Indeed, MAOA inhibition property was proved essential for chemotherapeutics to exhibit their substantial benefits in PCa [20]. Hence, MAO inhibition and inhibitors need intensive investigations.

For the complementary treatment of both PCa and depression, we aimed for a single molecule-based therapy inspired by versatile natural products. The polyphenolic antioxidant flavonoids possess various properties against many diseases, including neurodegeneration and cancer [28]. The recent association of flavonoids intake with lower PCa incidence and mortality signified their preventive and therapeutic properties [29]. Previous studies support that flavonoids with multidrug resistance (MDR) inhibitory properties in cancer embody a promising strategy for future chemopreventive therapy [30]. The flavonoids include flavones and chalcones subclasses. While most chalcones exhibit potent reversible MAO inhibitory effects [31], their scaffolds might possess antioxidant, cytotoxic, and apoptotic properties that were reported to hold promises as anti-cancer drugs [32,33]. Additionally, flavone scaffolds are well-known for drug development in inflammation, osteoporosis, cognitive dysfunction, and cardiovascular diseases [34]. More importantly, some flavone structures possess safe anticancer properties [35], inhibit MAO isozymes activities, and increase monoamine neurotransmitters synthesis [36]. Clinically, despite the challenges to reflect the in vitro achievements, research is on the go for flavonoids such as quercetin, genistein, and epigallocatechin gallate for their anticancer effects. Indeed, flavonoid nutraceuticals combined with common chemotherapeutics for chemopreventive anticancer activities recently showed progressively promising results [37]. Hence, synthesizing flavonoid-related structures will provide potentially valuable treatments for PCa and depression.

In the current study, we identified synthetic MAO-AIs for possible therapeutic use in the management of PCa with depression. Additionally, to predict the possible side effects of having consequential excessive essential and dietary monoamines that MAO-AIs might cause, we determined their safety in PCa treatment by investigating the top MAO-AIs mode of inhibition. In our search for compounds with these multifaceted actions, we investigated 20 synthesized compounds (KKRs) (Table 1) against recombinant human MAO isozyme (*h*MAO-A and *h*MAO-B) activities. We evaluated their antiproliferative effects against three human PCa metastatic cell lines. Our main objective was to find a reversible—and thus—safe RIMA compound with antiproliferative and anti-metastatic effects against aggressive PCa cells, with an endeavor to understand the underlying molecular mode of inhibitions. With multiple various in vitro screens conducted, we evaluate whether all MAO-Is can have anti-PCa effects.

## 2. Results

### 2.1. KKRs Inhibited Monoamine Oxidase-A

#### 2.1.1. KKRs High Throughput Screening for MAO-A Selective Inhibition

The inhibitory potentials of the twenty synthesized KKRs (Table 1) on A and B isozymes were tested and compared to different standard MAO-AIs and MAO-BIs, using MAO-luminescence assay (Figure 1). At first glance, the KKRs inhibited *h*MAO-A more than *h*MAO-B; all KKR IC_90_s against *h*MAO-B were more than 100 μM, which indicated a lower inhibitory potency than that of *h*MAO-A. The number and effectiveness of KKRs that inhibited *h*MAO-A were more than those that inhibited *h*MAO-B. Notably, five compounds out of the twenty KKRs (KKR11, KKR2A, KKR7, KKR20, and KKR21; Figure 1 dotted rectangle) exhibited MAO-A inhibitory effects matching the five tested standard MAO-Is (safinamide (SAF), pargyline (PARG), pirlindole (PIRL), (*R*)-(−)-deprenyl HCl (DEP), and clorgyline (CLORG)). The five KKRs affected the *h*MAO-A activity with an expected high potency (IC_90_ < 100 μM).

Comparing SAR of the top active five structures to SAR of their lesser non-active KKR analogs, we identified some specific functional groups in each active KKR that were crucial for causing or increasing the MAO inhibitory effects. By comparing the KKRs in Table 1 to their results in Figure 1, we observed the following—the chalcone KKR11 became more active when the fluorine atoms at the positions C_3′_ and C_4′_ of the B-ring of the less active analog KKR14 were substituted with the chlorine atoms; chalcone KKR20 gained more activity against *h*MAO-A and *h*MAO-B when the chlorine atom at para-position of the A-ring was removed to the meta-position in the less active analog KKR23; the activity of the chromone KKR2A was higher than its analog KKR16 activity that contained three extra methoxy-groups at its phenyl ethylene group; and flavone KKR7 activity increased dramatically in the absence of the methoxy-groups at C_4′_ and C_3′_ at the B-rings of its analog KKR5 and KKR9. Even with the lack of its C_3_-OH group at the C-ring, the effect of KKR7 against *h*MAO-A was preserved in its analog KKR21. Thus, further studies were carried out on these active compounds.

#### 2.1.2. The Most Inhibitory Potency of KKR Compounds against *h*MAO-A

The five top-ranked compounds against *h*MAO-A activity were further evaluated for their IC_50_ of *h*MAO-A and *h*MAO-B to determine the *h*MAO-A inhibitory potency and relative selectivity (RS_A_) (Figure 2). The specific inhibitory potencies were determined using the MAO-luminescence assay and compared to the standard, the selective MAO-AI PIRL. All tested compounds showed high-to-moderate MAO-A inhibition potencies (IC_50_ range 0.02–16 μM). The outstanding potency of flavone KKR21 against MAO-A (IC_50_ = 0.019 μM) matched the standard PIRL, the chalcone KKR11, and the chromone KKR2A (*p* > 0.05) (Figure 2a–d). Meanwhile, KKR21 potency was significantly higher than its analog KKR7 and chalcone KKR20 against MAO-A (*p* < 0.0001 and <0.001, respectively; Figure 2f,e).

In terms of selecting MAO-A inhibition, however, both flavones KKR21 and its analog KKR7 were highly selective MAO-AIs (RS_A_ > 42105-folds, *p* < 0.0001), matching the standard PIRL selectivity (Figure 2b). Notably, lacking the C_3_-OH group in KKR21, compared to KKR7, indicates the group implication in the higher potency of KKR21 against MAO-A without affecting selectivity. On the other hand, the top chalcones were less selective against MAO-A; chalcone KKR20 selectivity was in favor to inhibit MAO-B (RS_B_; *p* < 0.001) while chalcone KKR11 was a non-selective MAO-I (RS_A_
*p* > 0.05). Additionally, the differently structured chromone KKR2A showed a non-selective MAO inhibitory activity.

### 2.2. Mode of MAO-A Inhibition of KKR21 and KKR11

#### 2.2.1. Effects on *h*MAO-A Michaelis–Menten Parameters

To determine whether the KKR21 and KKR11 inhibitory effects of MAO-A activity cause the cheese-effect, we investigated their mode of inhibition through their effect on the Michaelis–Menten kinetics parameters. We illustrated our results as Lineweaver–Burk (LWB) linear regressions plot in the presence of luciferin derivative substrate (LDS) (Figure 3a,b), with V_max_, LDS K_m_, and alpha parameter changes in the presence of each KKR (Table 2).

At first glance on LWB, the data showed that the lines of increasing concentrations of each compound co-intersected at the Y-axis, which indicated a competitive mode of inhibition (Figure 3a,b). Compared to the control, KKR21 and KKR11 showed no significant change on *h*MAO-A V_max_, while LDS K_m_ values increased significantly with increase in each KKR concentration (reaching *p* < 0.0001). The X- and Y-intercept behaviors that reflected how the V_max_ and K_m_ were affected indicated that both KKRs competed with the substrate to inhibit the MAO-A isozyme.

As a further step to verify the competitiveness of KKR21 and KKR11 for *h*MAO-A inhibition, and consequently determine their K_i_s, we analyzed the Michaelis–Menten data for the best-fit enzyme inhibitory models using the GraphPad Prism. Via the mixed model for enzyme inhibition, we defined its alpha parameter value range (Table 2). Alpha parameter, the indicator for a mode of inhibition, designated a competitive mode for the *h*MAO-A inhibition (alpha larger than one) for both KKRs; alpha analysis rejected the non-competitive and the uncompetitive behaviors in both KKRs. Thus, the model evidenced that these two KKRs impeded substrate-binding affinity to *h*MAO-A.

The results obtained under the tested conditions supported the KKR21 and KKR11 reversible competitive inhibition of *h*MAO-A. Then, we determined the K_i_ values using Cheng–Prosuff’s equation for both KKRs. The K_i_ values were found to be small for KKR21 (1.7 nM) and KKR11 (7.4 nM) (Table 2). Since the lower the K_i_, the lower the KKR concentration needed to reduce the *h*MAO-A activity rate, we presumed that both KKRs possessed strong affinities to bind to the *h*MAO-A active site.

#### 2.2.2. Molecular Docking at the MAO-A Active Site

Docking studies were conducted to rationalize the molecular mechanisms of the reversible mode of inhibition of KKR21 and KKR11. The responsible interactions of the molecules with the crystal structure of the human MAO-A monopartite active site were predicted (Figure 4). Both KKRs of interest were well-docked to the narrow active site at which the standard harmine (2Z5X) interacted (Figure 4a,b). Our results indicated that the molecular shape and the dominant hydrophobic interactions at the active site were the main contributors to the high affinity and reversible competitiveness of the KKR ligands. The lowest free energy of the binding of KKR21 and KKR11 molecules were close to the standard (shape scores of −13.37 and −10.56, respectively) (Figure 4a). The phenyl rings of the open- and closed-chromone core structures of KKR11 and KKR21 were embedded at the site of the lipophilic zone between the entrance cavity and the MAO-cofactor flavin adenine dinucleotide (FAD) (Figure 4a,b—brown zone and blue ring, respectively). Our data indicated that these lipophilic interactions might have been initiated by the van der Waals bonds between the lipophilic KKR structures and the lipophilic zone of amino acid residues. The lipophilic KKR structures included the KKR21 closed-chromone core scaffold, the KKR21 C_2_-phenyl ring, the KKR11 open aliphatic chain scaffold, and the two KKR11 phenyl rings.

At the MAO-A active site, the standard 2Z5X and both KKRs shared all lipophilic interactions of the 17 specific mostly lipophilic amino acid residues, and five active interactions of water molecules (Table 3). Only the hydrophilic threonine-336A residue was an exception to interaction with the standard. Alternatively, both KKRs had hydrophilic interactions with asparagine 181A, FAD, and the active water-710 molecule, possibly because of the non-covalent ionic bonds with A-phenyl ring-substituted groups of the flavonoids, all of which were close to the FAD. Additional KKR11 interactions to lipophilic alanine-111 and leucine-97 were predicted. The chloride groups of the long KKR11 molecule might enhance these interactions at the lipophilic entrance of the protein. Interestingly, no significant predictions of H-bond interactions were involved; only KKR11 worked as a donor for an H-bond formed with tyrosine-444A, which might add a minor reversible competitive behavior in inhibiting the enzyme. These types of non-covalent interactions that were mostly shared with the standard might explain reversibility; reflect the high affinity, and the competitiveness of their mode of human MAO-A inhibition.

### 2.3. KKRs Cytotoxic and Antiproliferative Effects in PCa Cells

#### 2.3.1. Standards Effects on PCa Cells Proliferation

In the search for a comparative clinical standard that has anti-PCa effects, we tested three different human PCa cell lines (LNCaP, DU145, and PC3) for their sensitivity toward different clinical standards. The standards included two anti-PCa drugs, FLUT (a classical, non-steroidal, anti-androgen that belong to ADT drugs and is used for CRPC) and TAX (a generic chemotherapeutic drug), and two differently selective MAO-Is, PIRL (RIMA antidepressant drug), and DEP (drug recently used in depression [26]). Our preliminary data showed that the LNCaP cell line proliferation was the most sensitive, while PC3 resisted all standards (data not shown). Even for the LNCaP cells, none of the standards showed a significant antiproliferative effect at 10 μM, except TAX, which was highly effective even at 0.5 μM (*p* < 0.0001) (Figure 5a). Indeed, TAX had a very potent antiproliferative effect on LNCaP with a determined IC_50_ of 0.21 μM (Figure 5b). Therefore, the TAX standard was used for comparison with all tested KKRs.

#### 2.3.2. KKR Structures Affected the LNCaP Cells the Most

To identify the KKR compounds with antiproliferative effects, we conducted other high throughput screenings on the cellular level. We ranked the viability and proliferation of three different types of PCa cell lines in the presence of the twenty synthetic KKRs, ascendingly (Figure 6). The clinical drug, TAX, was used as a positive antiproliferative standard. The preliminary results showed that PC3 cells were the most resistant to KKRs and TAX. However, among the three cell lines and in comparison to PC3, KKRs affected the viability and slowed the proliferation rate of the LNCaP cells the most, followed by the DU145 group (*p* < 0.0001).

For the proliferation screen, KKRs were effective against LNCaP more than DU145, similarly to TAX., in LNCaP group, the top five KKRs that kept standing out with antiproliferative effects were chalcones; KKR11, 17, 18, 23, and KKR20; followed by the flavone KKR7 (Figure 6a). On the molecular level, these synthetics, except KKR17 that showed an entirely different behavior, contained chlorine atom substitutions at their chalcone structures, which may indicate chlorine essentiality against LNCaP in particular. Moreover, the top effective KKRs in the more resistant DU145 group are chalcones—KKR17 and KKR11—and the flavone KKR7 (Figure 6a: circled data points). In the context of MAO inhibition, it is worth noticing that three (KKR 11, 7, and 20) out of these top six KKRs were also among the top five KKRs of MAO-Is screenings (KKR 21, 20, 11, 7, and 2A), regardless of their remarkably different selectivities (Figure 2). These initial results indicate possible distinctive antiproliferative mechanisms of KKRs with and without MAO inhibitory properties.

In comparison to KKRs structure–activity relationship and as changing the analog substitution groups could affect the bioactive properties, some substitutions stood out. The chalcone KKR17 structure was effective against all investigated cell lines, including the metastatic PC3. Nevertheless, KKR17 did not have MAO inhibitory properties in our MAO-screens, which indicated the involvement of a different mechanism of action. However, when comparing the KKR17 with its analog KKR11 structure (Table 1), it became clear that the substitution of KKR11 chlorine atoms at the B-ring with the methoxy-groups to make KKR17 has dramatically switched the MAO inhibitory property of KKR11 to the PCa antiproliferative property of KKR17, which lost the MAO inhibitory property. Likewise, the antiproliferative chalcone KKR18, an analog of KKR20 (MAO-AI), points to the role of its dioxolane fused ring in the elimination of MAO-A inhibition property in preserving the anti-LNCaP effects. Meanwhile, the chorine para-position of KKR23 did not change the anti-PCa properties compared to KKR20. These PCa screening results highlighted compounds worthy of further considerations, particularly KKR11, 7, 20, and KKR17. On the other hand, KKR21, and KKR2A, which are among the top five MAO-A inhibitors, showed no effects on any of the screened PCa cell models, which might indicate resistance.

Assuming 90% viability as the margin of cytotoxicity (Figure 6b), the results showed that the vast majority of KKRs are non-toxic to the three tested cell lines up to 50 μM. Exceptions were four KKRs that affected the LNCaP cells (KKR17, KKR22, KKR16, and KKR7). The viability of the other two types of cells was not affected. The non-toxic behavior of KKRs with *h*MAO-A inhibitory properties was similar to TAX. Notably, KKR11 that showed safe inhibitions on both MAOs showed the least toxicity but with high antiproliferative effects on the two PCa cell lines. This behavior highlighted the compound as a great candidate for further studies.

#### 2.3.3. Three MAO-A Inhibiting KKRs Affected LNCaP Cells Proliferation Equally Potently

For the LNCaP screen results validation, MAO-Is KKR11, 7, and 20 were further investigated for their antiproliferative potency against LNCaP and were compared with KKR21 and 2A, along with the standard RIMA PIRL. Additionally, the non-MAO-I KKR17 potencies against the three PCa cells were included in the comparison (Figure 7).

The changes in the three cell lines growth were measured after a 72-h incubation period. As in the results, KKR11, KKR20, and KKR7 showed a superior antiproliferative effect against the LNCaP cells with insignificant differences between their potencies (IC_50_s of 9.5, 9.2, and 9.1 μM, respectively; *p* > 0.05) (Figure 7a). Meanwhile, the potencies of KKR21 (the KKR7 without C_3_-OH group) and KKR2A (similarly to PIRL) were much weaker—if existent—in the tested range of concentrations. These results indicated that not all MAO-AIs can work against PCa. More importantly, the C_3_-OH group of KKR7 that was responsible for weakening the MAO-A inhibitory potency appeared responsible for the cytotoxic and antiproliferative effects against the LNCaP cells, which indicated that the substituted groups of the flavone play a substantial role in determining the effect against LNCaP, independent from that of inhibiting MAO-A.

To validate the KKR17 exceptional effects, we determined its potencies against all tested PCa cell lines. Indeed, KKR17, which stood out in all our PCa cell screens, showed a different behavior from the top active three KKRs against PCa cells (Figure 7b). KKR17, which was inactive against MAOs, was the most potent against DU145 and moderately affected the PC3 proliferation (*p* < 0.0001), in addition to affecting the LNCaP. The findings indicated that the KKR17 anti-proliferation mechanism that affected all tested PCa cells was independent of any MAO inhibition.

#### 2.3.4. Three MAO-A Inhibiting KKRs Differently Affected the LNCaP Cells Morphology

The LNCaP cell morphological alterations, along with their proliferative growth, were monitored for several days of exposure to the five selected KKRs and PIRL. Fixed 20 μM concentrations were chosen to guarantee higher concentration than the IC_50_ of any tested compound and to ease the comparison between the compound effects. The morphological effects of the selected KKRs were highly pronounced after five-days of exposure (Figure 8).

Comparable to the proliferating control cells (Figure 8a), the KKR2A, KKR21, and PIRL had neither significant effects on LNCaP proliferation nor apparent changes in cell morphology (Figure 8b; right column of images). As shown in the left column of images, however, the KKR11, KKR20, and KKR7 had dramatically and differently altered cell morphology with a highly significant loss of proliferation (*p* < 0.0001). Impressively, with KKR11, the persisted LNCaP cells acquired damaged or undefined shape with an extended time of incubation. Meanwhile, KKR20 caused the cells to lose their neurite elongations to be spherical. Interestingly, KKR7 showed a differently irregular shape, and neurite elongations were observable. These differently induced morphological changes and damaging cancer cells indicated that each of the different structures of MAO-AIs affected the LNCaP in a distinctively structure-related mechanism.

#### 2.3.5. KKR11 Inhibited the DU145 Cells Migration

Our observations of the distinctive morphological alteration patterns to the LNCaP cells in Figure 8 encouraged further investigations regarding metastasis. We tested the ability of the three active KKRs (11, 7, and 20) to limit cell migration from the tumor site and whether they differ in their actions. We performed a commonly used scratch assay on the metastatic DU145 cells. While KKR20 and KKR7 showed no significant effects against cell migration (data are not shown), the preliminary data showed that only KKR11 and the positive control TAX delayed cell migration. Accordingly, KKR11 serial concentrations were tested on the cells for confirmed results (Figure 9).

After 72 h of incubation, the gap in the control was fully closed, and no sign of scratch was observed in most wells. Meanwhile, the scratch width of the gap (between the red lines) increased with increased KKR concentration; the KKR11 treated cells showed slower gap closure in a concentration-dependent manner. All imaged cells around the gap absorbed the blue stain and kept attached to the surface, indicating that the DU145 cells were viable after 72 h. From these data, we suggest that the metastatic cells lost their ability to migrate to the vacant space to close the gap. Thus, KKR11 treatment could have the potential to cause a dramatic failure of aggressive metastatic PCa cells, such as DU145, to migrate from the tumor site and invade the surrounding healthy tissues.

## 3. Discussion

The association between advanced PCa and vulnerability to depression among men, is becoming evident. Possible main contributors include the frustrating PCa symptoms and the depressive side effects of the current ADT therapy [12]. Recent reports associated PCa aggressiveness and metastasis with the overexpression of the MAO-A, the isozyme that was previously associated with depression [14,15]. Fortunately, studies on flavonoid compounds held promises as alternative therapies for their neuroprotective and chemoprotective properties. Therefore, we tested 20 synthetic flavonoids (KKRs; Table 1) for PCa and depression preventative activities by targeting PCa cell proliferation and selective MAO-A activity. The obtained data from our various screenings revealed that these lipophilic KKRs embraced single- and dual-functional characters regarding the effects on PCa and depression, by inhibiting MAOs and reducing PCa proliferation. Remarkably, the chalcone KKR11 was a competitive non-selective MAO-AI with PCa antiproliferative and anti-metastatic activities, making it an exceptional potential candidate for PCa patients with depression. On the contrary, the flavone KKR21 was a competitive and selective MAO-AI with no effects on PCa cells, making it comparable to the clinical standard PIRL used for depression. Meanwhile, KKR17, which was inactive against any MAO isozymes, slowed the proliferation rate of all tested PCa cell lines. Accordingly, KKR11 might act with a novel strategy for metastatic PCa patients with depression symptoms. These different properties of KKRs introduce drugs for PCa with antidepressant activities rather than using MAO-AIs only for depression and using the chemotherapeutics only for PCa.

In our high throughput screenings, the synthetic KKR flavonoids displayed a tendency to inhibit MAO-A potently and selectively (Figure 1), while more specifically showing antiproliferative effects on the PCa cells of LNCaP (Figure 6). Consistently, many natural flavonoids showed MAO inhibitory properties [38,39,40,41]. Moreover, previous synthetic substituted flavonoids exhibited antiproliferative effects on PCa [42]. Other reports of screened natural flavonoids showed antiproliferative effects with patterns of PCa cell-cycle arrest in LNCaP and PC3 cell lines [43]. The possible mechanisms of action of flavonoids in cancer chemoprevention and chemotherapy are various [44]. Those mechanisms included antiproliferative effects and regulating-enzymes for cell proliferation, inhibition of oxidative stress and pro-oxidant processes, pro-apoptotic mechanisms, and reversal of chemo-resistance [45,46,47]. Consistent with the previous findings, the obtained data from our various screens indicated that KKR structures embrace multifunctional candidate drugs for managing PCa with depression, possibly through the mechanism of inhibiting MAOs activity and reducing PCa proliferation.

The effects of the top KKRs on *h*MAO-A and PCa models screens are widely intersected (Figure 5, Figure 7 and Figure 8). The finding that three out of the top six antiproliferative KKRs were MAO-Is (KKR11, KKR7, and KKR20) supports the previously reported MAO-A association and responsibility for aggravated PCa [17,18,19,20]. These MAO-AIs hold promises as new therapeutic options for PCa patients. Indeed, by inhibiting the chronically hyperactive MAO-A and, as previously proposed [48], reducing enzyme-related oxidative stress production in PCa cells, MAO-AIs could manage malignancy and aggressiveness. With a further mechanism of inhibiting MAO-B, they can manage depressive symptoms in men. The clinical and pharmacological MAO-Is increased the neurotransmitters in the nervous system, reduced H_2_O_2_ and oxidative stress, and provided neuroprotection [49], without which depression patients on other treatments would have MAO-A toxic activity that would remain aggravated [50]. However, although these three KKRs share MAO-A inhibition, their varied effects on PCa morphology and migration might not limit their mechanism to inhibiting the enzymes; rather presume additional mechanisms involved in affecting cell proliferation, besides enzyme inhibition.

Other top KKRs were of a single effect; MAO-I or anti-proliferative. The KKR17 recurrent effects on all tested PCa cells (Figure 7) might indicate a structurally related different mechanism of action against PCa cells that did not involve MAO inhibition. For its similarity to KKR11—the most active KKR in this work—we related its different anti-PCa-function to the substituted methoxy-groups in its structure instead of the chlorine atoms. For its relatively unique character, KKR17 is worthy of being considered for more in-depth future investigations, for PCa management. Regarding KKR21 and KKR2A, however, and despite their potent MAO-A inhibitory effects (Figure 2), they were not able to affect any PCa cell lines (Figure 6, Figure 7 and Figure 8). Likewise, some standard MAO-Is showed no effects on PCa cell proliferation (Figure 5). The well-known RIMA standard PIRL did not have any detectable effects on viability, proliferation, nor changes in cell shape (Figure 5, Figure 7 and Figure 8). We presumed that these MAO-Is were unable to have access to the cancer cells and reach PCa cellular MAOs, or were inactivated by enzymatic metabolism in the cancer cell. However, future independent investigations should be directed to address inefficacy. Nonetheless, these two compounds should also be considered for further neurodegeneration investigations for depression.

It is interesting that MAO-Is KKRs did not have any detectable effects on PC3, while the only compound that affected the PC3 cells, chalcone KKR17, was not an MAO-I, and yet, PC3 continued to be the most resistant. All PCa cells under investigation were epithelial cell lines that expressed MAO-A and not MAO-B [20]. By looking at the MAO-A levels in these cell lines we found that the cellular phenotype LNCaP expressed both MAO-A transcript and protein in much greater levels than that the DU145 and PC3 cells [20]. This might explain LNCaP susceptibility to a higher number and high potency of KKRs compared to the DU145 and PC3 cells. Conversely, the low MAO expressions of PC3 cells might increase resistance to KKRs in this aspect. Nevertheless, the MAO-independent antiapoptotic effects of KKR17, once again, approved a different mechanism of affecting the three cell lines, regardless of their MAO expression or activity.

With their low MAO-A expression, PC3 cells might possess other characteristics that increased their resistance to KKRs in general. PC3 cells exhibit characteristics of highly malignant but rare prostatic small cell carcinoma. This differs from LNCaP, which represents the common prostatic adenocarcinoma characters [51]. Indeed, there are characters of PC3 cells that increase its resistance compared to the LNCaP cells, such as the positivity for the stem cell-associated marker CD44, which enables PC3 cells for better intercellular interactions and migration [51]. More importantly, only PC3 cells have low levels of topoisomerase 2α (TOP2α), the double-strand breaker that has been associated with chemotherapy resistance [52]. While KKR17 has demonstrated great potentials as a cytotoxic anti-proliferative agent in all three PCa phenotypes but with the least effects on PC3, the highest potency of KKR17 was against DU145 cells.

Contrary to PC3, the fast proliferative DU145 cells expressed the highest levels of TOP2α, which is considered a target for the main anticancer drugs [53] and other versatile catalytic TOP2α inhibitors [54]. Indeed, KKR17 dose-dependent inhibition of DU145 cell proliferation displayed a sharp slope with an angle that varied from the LNCaP and PC3 slopes of inhibition (Figure 7b), which hypothetically reflects its proportion to TOP2α. Regardless, several flavonoids were previously reported to be effective against DU145, such as genistein, apigenin, chrysin, and quercetin [55]. As a flavonoid derivative, KKR17 might have promoted apoptosis, necrosis, cell cycle arrest, or autophagy to inhibit the proliferative growth of cells, particularly, the androgen-independent DU145 phenotype, which should be further investigated.

Predominantly, KKR11 kept standing out in all of our investigations. KKR11 potently inhibited MAO-A and MAO-B activities, showed reversibility of MAO-A inhibition with high affinity, markedly inhibited PCa cell growth, changed LNCaP morphology, and slowed DU145 migration (Table 1, Figure 4, Figure 5, Figure 6, Figure 7, Figure 8 and Figure 9). Consistent reports on synthetic standard MAO-Is indicated anti-PCa effects; the non-selective irreversible MAO-I phenelzine and MAO-BI PARG decreased LNCaP viability and inhibited their metastasis [27]. More reports were available on CLORG, another classical irreversible MAO-AI with a structure related to PARG. CORG inhibited the growth of androgen-independent metastatic VCaP xenograft cells in mice and many oncogenic pathways, and induced AR and genes associated with secretory cell differentiation [56]. Another report showed that CLORG enhanced differentiation in primary prostatic epithelial cells by inducing morphological changes that resembled secretory differentiation and ARs restoration [18]; the study suggested CLORG for patients with androgen-independent aggressive PCa and metastasis. Moreover, the combination of CLORG with docetaxel, a conventional chemotherapeutic drug used in PCa, limited in vitro PC3 and LNCaP cells resistance by increasing docetaxel anti-proliferative effects and apoptosis [20]. Other synthesized derivatives of 2-imidazoline, pyrrolo[3,4-f]indole-5,7-dione, and indole-5,6-dicarbonitrile were reported as MAO-Is and were suggested for the treatment of different diseases, including depression or PCa [57,58]. In a comparison of KKR11 to other flavonoid structures, specific natural flavonoids were found to share multiple actions related to PCa or depression in a variety of studies. An example is genistein, a soybean isoflavone, which proved to reverse MAO-A inhibition [41], improved cognitive functions in males [59], and exhibited antioxidant and anti-cancer effects [60,61]. In PCa cells, genistein protected from aging-related DNA damage [62], inhibited epithelial-to-mesenchymal transition [63], and LNCaP invasiveness [64]. Consequently, KKR11 promises to be a beneficial treatment for metastatic aggressive PCa, which is concurrent with depression. MAO-AIs potential to eliminate metastasis grants them to be novel therapeutics for PCa and depression patients, which needs to be emphasized.

Our kinetics results for an *h*MAO-A mode of inhibition and docking studies indicated KKR11 and KKR21 strong hydrophobic binding affinities with competitive modes of inhibition (Figure 3 and Figure 4). In other words, the results showed secure reversibility [65]. Recent research recognized the importance of MAO-AIs reversibility for safer medication and recommended using RIMA [66]. In the past, rare incidences of hypertensive-crisis were reported after the ingestion of dairy products containing the monoamine tyramine, a specific MAO-A substrate in the intestine [67]; the irreversible inhibitory mechanism of many MAO-AIs limit their use. Nonetheless, the use of reversible MAO-AIs prevent this problem by allowing tyramine to displace the inhibitor of intestinal MAO-A to be metabolized and preventing hypertensive crisis [21]. For the establishment of better selective, reversible, and potent molecules against MAO, a previous in-silico docking survey proposed modified natural flavonoid scaffolds as safe candidates [68], and a new generation of privileged scaffolds as highly selective and reversible MAO-Is have revived the interest [69]. Currently, the therapeutic value of the RIMA drugs for depression is a well-established strategy [70]. The nonselective-dose with a reversible MAO-A inhibitory mode of the current MAO-BI, deprenyl (DEP), was decisively proven for elderly patients with depression (Emsam^®^) [26]. However, DEP has no effects on PCa cells (Figure 5). Further, some natural flavonoids were recently reported to have reversible MAO mode of inhibition [38].

Here, regardless of their MAO-B inhibition, we reported two synthetic flavonoid derivatives with potential reversible MAO-A inhibitory properties, KKR21 and KKR11, which indicate their safer use on patients. Moreover, KKR11 would have the advantage of safe reversibility compared to CLORG or other previously reported MAO-Is with PCa effects. Hypothetically, the tight but reversible binding of KKR11 and KKR21 to MAO-A is expected to allow the recovery of MAO-A activity after their in-vivo withdrawal or in the presence of overly high tyramine and other essential monoamine neurotransmitters levels, which would allow them to avoid consequential adverse effects [71]. When tyramine metabolism is secured, non-selective KKR11 will also have the advantage of inhibiting MAO-B cytotoxic byproducts responsible for oxidative stress. On the other hand, the higher affinity of MAO-A KKR21 as compared to KKR11, the selectivity to only inhibit *h*MAO-A, and the potency that is comparable to standard RIMA, PIRL (Figure 2, Figure 3 and Figure 4) provides further reasons for investigating the use of KKR21 specifically for depressed patients.

The predicted hydrophobic interactions of KKR11 and KKR21 in our docking studies at the MAO-A active site (Figure 4) matched previous predictions reported on other docked flavonoids with MAO reversible inhibitory effects or RIMA. These predictions might explain the high affinity and inhibition reversibility of the two compounds toward the enzyme. It is well-known that the human MAO-A active site consists of a hydrophobic cavity lined by characteristic residues located in front of the FAD ring and extended to the enzyme surface. In front of the FAD, the two well-known interactive residues of tyrosine-407/tyrosine-444 aromatic sandwich stabilize the substrate binding [72]. Our predicted reversible interactions of KKR21, KKR11, and the standard 2Z5X with the aromatic sandwich and other several amino acid residues support the impeding of substrates binding. Moreover, the gating pair of phenylalanine-208/isoleucine-335 at the active site has no match in MAO-B [73]. This exclusive interaction determines the different properties of the MAO-A substrate and inhibitor [72]. Thus, our predicted interactions of the two KKRs with isoleucine-335 can explain their role in MAO-A potent inhibition, in addition to increased KKR21 selectivity.

On the other hand, extra hydrophilic interactions of KKR21 and KKR11 with asparagine-181 and FAD were not predicted for 2Z5X (Table 3). However, the interactions with asparagine-181 were previously predicted with MAO-A inhibiting flavonoids of quercetin and kaempferol-3-monoglucoside [39]. Additionally, consistent to KKRs interactions with water molecules, our docking already found predicted reversible interactions of FAD and water with the MAO-A inhibiting isoflavone biochanin-A [40]. Similarly, additional predicted KKR11 interactions to alanine-111 residue are consistent with the reported flavanone naringenin docking interactions [74]. However, KKR11 interaction with leucine-97 might need more reports to support it and drive attention to its possible role in influencing non-selective KKR11 inhibitory potency, in comparison to KKR21. The consistency of our docking data with the previous reports highlights the critical role of these non-covalent interactions in the inhibitory potency, selectivity, and reversibility of KKR21, KKR11, and, generally, in RIMA flavonoids.

## 4. Materials and Methods

### 4.1. Materials and Synthesis

#### 4.1.1. Materials

The reagent kits of MAO-Glow™ were obtained from Promega Corporation (Madison, WI, USA). The recombinant human monoamine oxidase-A (*h*MAO-A) and -B (*h*MAO-B) derived from recombinant baculovirus were purchased from Sigma-Aldrich Co. (St. Louis, MO, USA). From the same company, we purchased chemicals including resazurin, TAX, FLUT, PIRL, DEP, SAF, PARG, and CLORG, and solutions, including Hank’s Balanced Salt Solution (HBSS), *N*-2-hydroxyethylpiperazine-*N*-ethanesulfonic acid buffer (HEPES), dimethyl sulfoxide (DMSO), and 0.25% Trypsin-EDTA solution (T/E).

Human PCa cell lines, including LNCaP, DU145, and PC3, derived from metastatic sites, were obtained from American Type Culture Collection (ATCC) (Manassas, VA, USA). Cell-culture flasks and other related equipment were obtained from Santa Cruz Biotechnology Inc. (Dallas, TX, USA). Dulbecco’s Phosphate Buffered Saline (PBS), 10,000 U/mL penicillin-G sodium/10 mg/mL streptomycin sulfate (P/S) were obtained from Atlanta Biologicals (Atlanta, GA, USA). Cell-culture media of RPMI1640, EMEM, DMEM/F12K gibco^®^, and fetal bovine serum (FBS), and other equipment were purchased from VWR Int. (Radnor, PA, USA).

#### 4.1.2. KKR Synthesis

The synthesis of substituted chalcone analogs was carried out via Claisen–Schmidt condensation, and flavone analogs were synthesized using a three-step Baker–Venkataraman rearrangement. Synthesis and characterization of these compounds were reported in our previous papers [75,76].

### 4.2. Investigating the Inhibition of Two Monoamine Oxidase Isozymes

#### 4.2.1. Isozymes Reactions Optimization

We optimized our control *h*MAO-A and *h*MAO-B isozymes reactions in the incubation condition, at 37 °C for 1 h, by performing a luminescence assay using the MAO-Glow™ kit [77]. Upon arrival, each isozyme was diluted with 10 mM HEPES in cold HBSS of pH 7.4. Afterward, the diluted isozymes were kept at −80 °C until use. Enzyme concentration optimization was accomplished by testing ten 2X serially diluted *h*MAO-A and *h*MAO-B concentrations for a final range of 7.00–0.01 U/mL each. The tested isozymes were incubated with LDS of 40 and 4 μM for isozymes A and B, respectively (recommended by Promega protocol), and a maximum of used DMSO of 1% final concentration. Different isozyme concentrations reactions at their initial velocities were monitored. Consequently, their required concentrations were defined to be 0.44 U/mL each. Additionally, ten different LDS concentrations, ranging from 0.3–150 μM, were monitored for optimizing and validating the linearity of the initial reaction for the *h*MAO-A mode of action. Subsequently, we chose a saturated substrate of 75 μM as a required concentration for the *h*MAO-A enzyme kinetic reactions.

#### 4.2.2. Screening KKR Compounds for Inhibiting *h*MAO-A and *h*MAO-B Activities

The inhibitory effects of the synthesized KKRs of *h*MAO-A and B were tested by measuring the decrease in arbitrary light units (ALU) signal produced by the optimized isozymes using the MAO-Glow™ Assay at 37 °C. In this assay, DMSO stock for each KKR compound was diluted with a reaction buffer of pH 7.4 to make a 4X working solution for a maximum final concentration of 100 μM and ≤1% DMSO. The experiments were run in the absence and presence of 4X the miscellaneous standards of MAO-BIs (DEP, PARG, and SAF) and MAO-AIs (PIRL and CLORG), with similar concentrations. In white opaque 96-well microplates, 12.5 μL of each compound was plated at RT and was followed by 25 μL of 2X cold *h*MAO-A or *h*MAO-B, for a final concentration of 0.44 U/mL. A related buffer substituted each isozyme to make the blank wells and substituted each compound to make the negative control wells. Following this, we added 12.5 μL of 4X LDS to make a final substrate concentration of 40- and 4-μM for *h*MAO-A and *h*MAO-B, respectively. All screen plates were incubated at 37 °C for 60 min. The plates of 50 μL/well were then taken out from the incubator, and their reactions were stopped by an equal volume of 50 μL of reporter luciferin detection reagent (RLDR). After 30 min, the luminescence signals for all plates were read on a Synergy HTX Multi-reader from Bio-Tek (Winooski, VT, USA).

#### 4.2.3. Relative Selectivity against *h*MAO-A

The *h*MAO-A and B inhibitory potencies and the relative selectivity (RS) of the most effective five KKRs of 2A, 7, 11, 20, and 21 against MAO-A were determined by using the same MAO-Glow™ assay at 37 °C. The experiments were performed in the presence and absence of the standard PILR. In white opaque 96-well plates, each KKR compound was serially diluted with a reaction buffer of pH 7.4 at the highest possible maximum final concentration, and the least used DMSO (<2%). In blank wells, the related reaction buffer substituted the isozyme. In negative control wells, the reaction buffer substituted each compound concentration. Then, the procedure was carried out as mentioned above (Section 4.2.2), and the reaction was detected on the Bio-Tek Synergy HTX Multi-Reader. The IC_50_ of each KKR on both isozymes was determined and subsequently the RS_A_ and RS_B_ was calculated.

#### 4.2.4. Michaelis–Menten Kinetics Determination

The *h*MAO-A modes of inhibition by KKR21 and KKR11 were investigated using the Michaelis–Menten reaction kinetics and MAO-Glow™ assay. A negative control buffer and seven serially diluted LDS concentrations were prepared from a maximum final of 75 μM. Other negative control buffers and three different concentrations of each KKR were prepared to resemble IC_50_/4, IC_50_, and IC_50_ × 4 final concentrations (final KKR11 of 0.08, 0.31, and 1.24 μM, and final KKR21 of 0.005, 0.019, and 0.076 μM, respectively). Each KKR was mixed with the isozyme in a 2:1 ratio volume for a final concentration of 0.44 U/mL. The reactions were initiated by adding 37.5 μL of the prepared mixture to 12.5 μL LDS substrate in the 96-well plates. Opaque plates were incubated at 37 °C for 60 min. The enzymatic reaction was terminated by adding 50 μL RLDR to the plates and then was re-incubated for 30 min at RT. The generated ALU was read on the Bio-Tek Synergy HTX Multi-Reader.

#### 4.2.5. Molecular Docking

An in-silico designed molecular structure of human MAO-A and the KKR molecular structures were used to recognize the possible interaction mechanisms with the isozyme active site. From the RCSB Protein Data Bank (RCSB PDB, www.rcsb.org), we downloaded the X-ray crystal structure of human MAO-A in complex with harmine (PDB ID: 2Z5X) and imported it into Sybyl-X program system (version 1.3) from Tripos International (St. Louis, MO, USA). The MAO-A’s chain-A was extracted and refined with the Biopolymer Structure Preparation Tool. All types of atoms were corrected, amino acid residues were repaired, and all hydrogen atoms were added to the structure before the docking. Additionally, MAO-A cofactor, flavin adenine dinucleotide (FAD), and its covalent linkage to the cysteine (CYS: 406: A) residue of the MAO-A and the active water molecules were retained during the docking process. This was followed by energy minimization of the MAO-A structure using the MMFF94 and MMFF94s force field charges.

For the method validation, the 2Z5X, as a bound standard ligand, was sketched using the Sybyl sketch, then re-docked to the crystal structure of the MAO-A; the value of root means square deviation (RMSD) between the retrieved and re-docked poses of the ligand was less than 2 Å. Next, for docking KKRs as test ligands, the low energy 3D conformers of each compound were generated using OMEGA (version 2.4.6) and docked using a dock type Hybrid of OEDocking (version 3.0.1) of OpenEye Scientific Software Incorporated (Santa Fe, NM, USA) [78]. We docked each KKR to the enzyme protein structure at its ligand binding domain. The top ten poses were considered, and the H-bonds formed between each KKR and the MAO-A amino acid residues of the top poses were measured.

### 4.3. Testing the Effects on Prostate Cancer Cell Proliferation and Metastasis

#### 4.3.1. Cell-Culture

Cell lines of LNCaP, DU145, and PC3 were cultured in complete cell-culture media and air-vented TC-flasks at 37 °C in humidified 5% CO_2_ incubator. The LNCaP cells were grown in RPMI1640 media, the DU145 cells in EMEM media, and the PC3 cells in DMEM/F12K media. All culture media were supplemented with 10% heat inactivated FBS and 1% P/S *v*/*v*. Cell media were maintained once to three times weekly. The three cell lines were routinely subcultured every three to six days, depending on the cell type, using 0.025% T/E for cell detachment.

#### 4.3.2. Screening for Effective KKRs against PCa-Cell Lines

##### The Viability Screen Assay

We examined the cytotoxic effects of all 20 synthesized KKRs on the three human PCa cell lines LNCaP, DU145, and PC3, after a 24 h exposure period. Non-toxic TAX was used for comparison. Briefly, cultured cells were seeded in black transparent bottom 96-well microplates at 50,000 cells/175-µL/well in experimental media (RPMI1640 for LNCaP, EMEM for DU145, and DMEM/F12K for PC3), supplied with 5% FBS and 1% P/S. Wells of media without cells were considered blanks and negative controls. The plates were incubated overnight at 37 °C in 5% CO_2_ to allow cell attachment. For each compound, a working solution of 8X KKR in sterile 10 μM HEPES in HBSS pH buffer of 7.4 was prepared for a final concentration of 50 μM each. The three attached cell types were treated immediately with the related buffer being added to the control wells. A sterile fluorescence-based solution of the resazurin reagent of 0.5 mg/mL in HBSS was prepared. After 24 h, the cells were exposed to 15% *v*/*v* of the sterile resazurin reagent, and the plates were re-incubated for 4.5 h to allow redox reaction. All plates were read under 530/590-nm excitation/emission wavelengths using Bio-Tek Synergy HTX Multi-reader as a microplate fluorescence reader.

##### The Antiproliferation Screen Assay

In this screen, we examined the growth inhibitory effects of all synthesized KKRs on the three human PCa cell lines LNCaP, DU145, and PC3. The antiproliferative clinical drug TAX of 0.25 μM was used as a positive control. Additionally, different standards of MAO-Is and FLUT were tested for their effect on the LNCaP cells. Briefly, the procedures and materials used were the same as for the viability assay except for the following—cultured cells were seeded at 10,000 cells/175-µL/well related experimental media, and the resazurin solution was added after 72 h of incubation with the compounds of 50 μM each.

#### 4.3.3. Proliferation and Morphological Observations of the PCa Cells with Top Effective KKRs

##### Antiproliferation Dose-Response Assay for 72 h

For the top effective KKRs, along with TAX, their potency to reduce the affected PCa cell line growth in a dose-dependent manner was determined using the resazurin fluorescence assay. We followed the same protocols and materials used for the antiproliferation screen assays (Section 4.3.2), with the following modifications. The maximum final KKR concentration used was 40 μM. The concentration of 8X of each KKR working solution in 1% HEPES 7.4 was serially diluted in a 3:1 ratio, compound to buffer, respectively. Each data set was collected from ten data points of 3.7–40 μM range. The control had the KKR substituted with buffer. The related IC_50_ averages were analyzed for the compounds that affected the cell lines the most.

##### Imaging the Proliferative LNCaP after Five-Days

The morphological alterations of the LNCaP cells were imaged, and their percentage proliferation rate was measured after five-days of incubation. The method of subculture and treatment were similar to our antiproliferation assay, with slight modifications. Briefly, the cells were seeded in transparent flat-bottom 96-well microplates with 10,000 cells/175 μL/well of RPMI1460 experimental media. Following overnight incubation, 20 μM concentration level (~2 × IC_50_) for each of KKR11, 20, 7, 2A, 21, and the standard PIRL were added to the cells for a total volume of 200 μL/well and re-incubated for five-days. On the day of imaging, 20X phase-contrast images of lid-covered treated and control cells were captured using an inverted microscope. Resazurin was then added for viability detection, and the imaged cells in the wells were read after 4.5 h on the Bio-Tek Synergy HTX Multi-reader.

#### 4.3.4. Migration Assay, Fixing, and Imaging DU145 Cells

For in-vitro migration experiments, the PCa metastatic DU145 cell line was used. Cell proliferation and cytotoxicity were monitored for assay validation. The cells (passage # ≤ 10) were grown in fresh media of EMEM 10% FBS, 1% P/S. The cells were seeded in flat-bottom transparent 96-well microplates (100,000 cells/100 μL/well) in EMEM 5% FBS, 1% P/S experimental media. The next day, the confluent cells monolayer was artificially injured by scratching a line across each well, using a 200 μL tip. Immediately, the old media with the detached cells were replaced with 175 μL fresh warm experimental media. Intact cells were then treated with 25 μL of HBSS buffer 7.4 containing tested KKR. In addition to the negative control, KKR11 was tested at gradual concentrations to make finals of 40, 30, 23, 17, and 13 μM. The cells were then incubated for an additional 72 h for gap closure of the control wells.

At the gap closure time point, all cells were subjected to the crystal violet assay that only stains viable cells. Briefly, all media were carefully aspired from cells that were then washed with Ca- and Mg-free PBS. The cells were then fixed with 100 μL 0.25% *v*/*v* glutaraldehyde, at 37 °C for 20–30 min incubation time, and dried overnight. The next day, all viable attached cells were stained with a 100 μL crystal violet solution (0.1% in DI-water) for up to 30 min. The solution was then discarded, and the stained cells were kept dry for imaging. Images of the scratched areas were photographed using 10X phase contrast objective lenses of Motic-MXH-100-AE31E-Elite Inverted Microscope.

### 4.4. Statistical Analyses

Statistical analyses were performed using the GraphPad Prism program (version 6.07 for Windows) from the GraphPad Prism Software Incorporated (San Diego, CA, USA). All data points and parameters were expressed as the mean ± SEM. All parameters were calculated from non-linear and linear regressions with the best R^2^ values, no less than 93%. The inhibitory concentration of 50% of isozyme (IC_50_) of each compound was obtained by the interpolation of a normalized variable slope logarithmic curve. The RS_A_ values of the selected compounds were calculated from their IC_50_ average means following the equation
RS_A_ = *h*MAO − B IC_50_/*H*mao − A IC_50_(1)

The V_max_ and K_m_ were calculated from the averaged X and Y intercepts of Lineweaver–Burk (LWB) linear regressions that were analyzed from the Michaelis–Menten curve data.
V_max_ = 1/Y-intercept, K_m_ = −1/X-intercept(2)

Inhibitor-constant (K_i_) was calculated from Cheng–Prosuff’s equation
K_i (comp)_ = IC_50_/(1+[S]/K_m_)(3)

The significance of differences among the groups of each data set was determined using one-way ANOVA, followed by Dunnett’s multiple comparisons test, and between different data sets using two-way ANOVA, followed by Sidak’s multiple comparisons test, or as it is mentioned for each figure; *p* > 0.05 was considered non-significant (ns).

## 5. Conclusions

In the current study, we established the basis for which the flavone, chalcone, and chromone synthetic structures might work as a treatment option for PCa patients with depression by reversibly inhibiting MAO-A and reducing oxidative stress. Our top compounds from MAO and anti-PCa screening were widely intersected. Our results were consistent with previous investigations on the discipline of PCa and depression. However, the results and SAR studies indicate that only certain MAO-AIs can have anti-PCa effects, and they do not have to be selective to MAO-A to exhibit the anti-PCa property. From the top five MAO-AIs and the top six anti-PCa proliferation determined, three top compounds intersected (KKR11, KKR7, and KKR20), two compounds worked only as MAO-Is without anti-PCa activity (KKR21 and KKR2A), and three affected PCa cells without MAO inhibitory property (KKR17, 18, and 23). Predominantly, KKR17 affected all tested PCa-cells proliferation rate without the MAO inhibition mechanism. We suggest the structure of methoxy-groups cruciality for these unique actions—and thus—proposed KKR17 as a candidate for further investigations on PCa and its mechanism. We also presumed that MAO-Is that did not affect PCa were unable to reach MAO-A, due to their structure or lack of essential active groups that are crucial for the multiple effects of the three intersected KKRs, such as the chlorine atoms and hydroxyl groups.

Further, the striking experimental reversibility and potency of both KKR21 and KKR11 possibly occur through tight non-covalent lipophilic interactions at the active site of MAO-A. Our molecular docking studies matched previous predictions on flavonoids with MAO reversible inhibitory mechanism and support the impeding of the binding substrates. The reversibility of inhibition was advantageous for its safe use without the serious side effect of the hypertensive crisis. Interactions of the two KKRs with isoleucine-335 explain their role in MAO-A potent inhibition. In contrast, KKR11 interaction with leucine-97 might need to drive more attention to its possible role in its non-selective inhibition. Nonetheless, KKR21 selectivity and high potency is comparable to the clinical RIMA drug PIRL, making it a potential antidepressant drug candidate with fewer side effects but not for PCa treatment.

More importantly, KKR11 is among the MAO-Is that share PCa antiproliferative effects on aggressive LNCaP and migrating metastatic DU145 cells. The potential non-selectivity and reversible MAO-A inhibitory property that reduces oxidative stress and impedes PCa malignancy promotion and aggressiveness is a possible mechanism. The KKR11 shape, lipophilicity, and the chlorine groups might play a significant role in these in vitro actions. Consistent results were found with clorgyline, pargyline, and phenelzine clinical and pharmacological drugs. However, KKR11 is superior for its reversible activity. The outstanding activities of KKR11 suggest a novel treatment for the management of the PCa disease, accompanied by depression. Strategies indicated might include single or adjuvant chemotherapy management. However, further studies are needed for developing the lead KKR compound. Additionally, for potential single-molecule based therapeutics, such as KKR11 for this domain, it is crucial to determine other possible in-vitro mechanisms of action and establish in-vivo evaluations in PCa models.

## Figures and Tables

**Figure 1 molecules-25-02257-f001:**
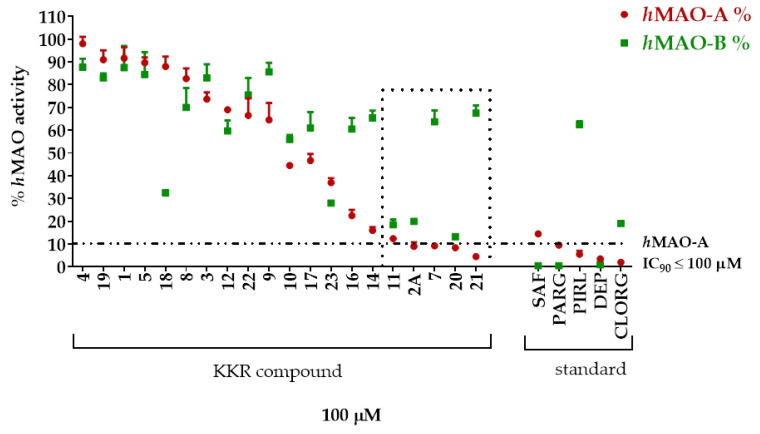
High throughput screening of KKR structures for top relative inhibitors of human monoamine oxidase-A (*h*MAO-A) (RI_A_). Synthetic compounds of 20 KKRs were tested at 100 μM for *h*MAO-A and *h*MAO-B inhibitory effects, and were compared to similar concentrations of standard MAO-AIs pirlindole (PIRL) and clorgyline (CLORG), and standard MAO-BIs safinamide (SAF), pargyline (PARG), and deprenyl (DEP). The descending order of inhibited *h*MAO-A activity revealed the top five active compounds (dotted rectangle) that inhibited > 90% of *h*MAO-A activity (dotted horizontal line).

**Figure 2 molecules-25-02257-f002:**
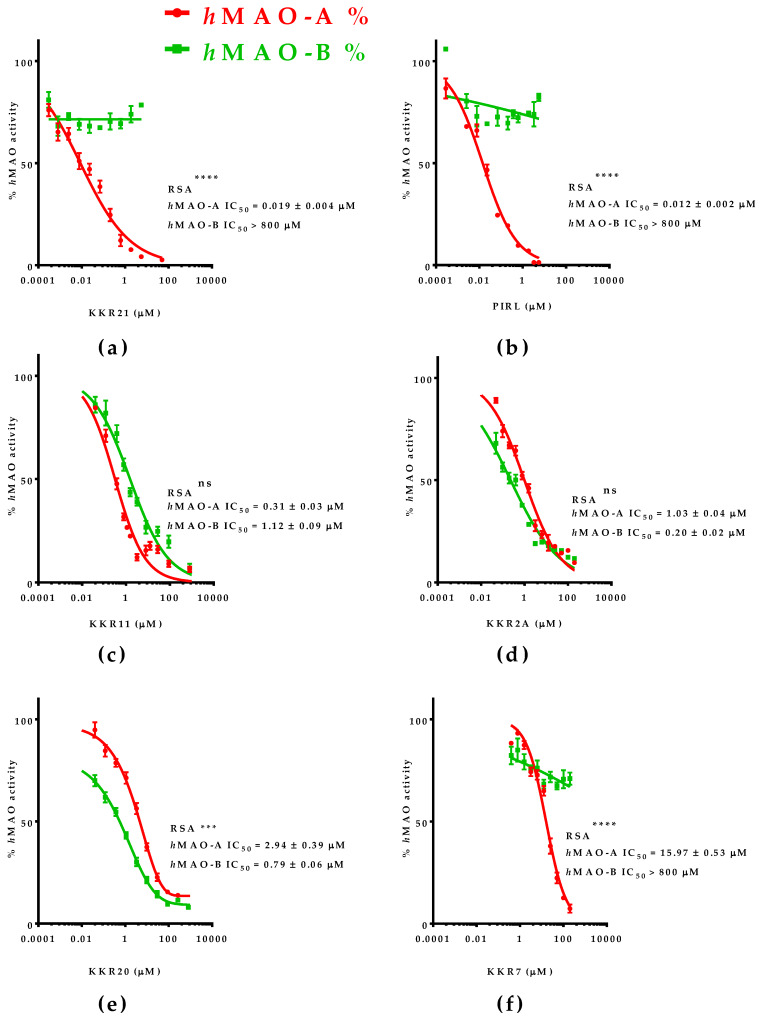
KKR *h*MAO-A and *h*MAO-B inhibitory potencies and *h*MAO-A relative selectivity (RS_A_); (**a**) KKR21, (**b**) standard PIRL, (**c**) KKR11, (**d**) KKR2A, (**e**) KKR20, and (**f**) KKR7. All compounds were equally potent MAO-AIs with different RS_A_. The percent points were presented as the mean ± SEM, *n* ≥ 4. The IC_50_ ± SEM values were averaged from at least two experiments. Significance of difference between the two isozymes IC_50_s for each compound was determined using two-way ANOVA, followed by Sidak’s multiple comparisons tests. ns *p* > 0.05, *** *p* < 0.001, **** *p* < 0.0001.

**Figure 3 molecules-25-02257-f003:**
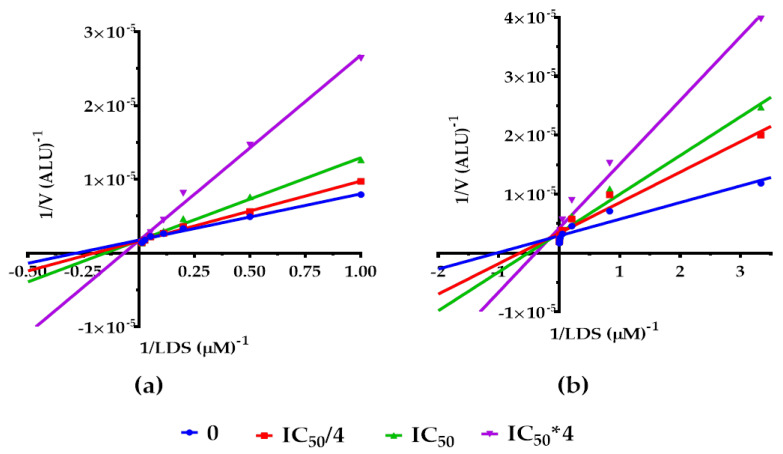
Michaelis–Menten kinetics of *h*MAO-A in the presence of KKR21 (**a**) and KKR11 (**b**). The representative Lineweaver–Burk (LWB) plots are of the inhibited *h*MAO-A with increasing luciferin derivative substrate (LDS) at the initial velocity (V) of producing arbitrary light units (ALU) at 37 °C. Three inhibitor concentrations were used; IC_50_/4, IC_50_, IC_50_ × 4. Data points were expressed as the mean ± SEM, representing more than two experiments with *n* = 3 each.

**Figure 4 molecules-25-02257-f004:**
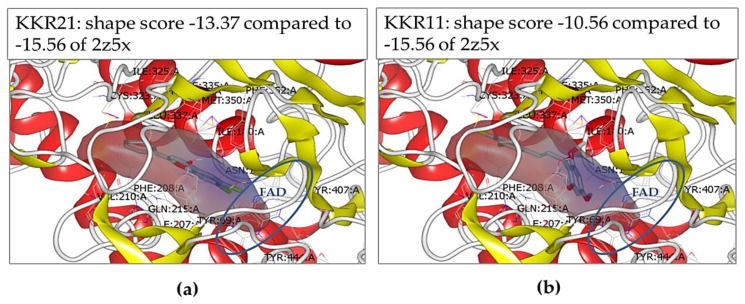
Docking conformation within human MAO-A crystal structure active site; docked KKR21 (**a**) and KKR11 (**b**) are the gray molecules. The active site zones include the lipophilic zones (brown), the hydrophilic zones (blue), and the neutral zones (gray). The cofactor flavin adenine dinucleotide (FAD) is in the blue ring.

**Figure 5 molecules-25-02257-f005:**
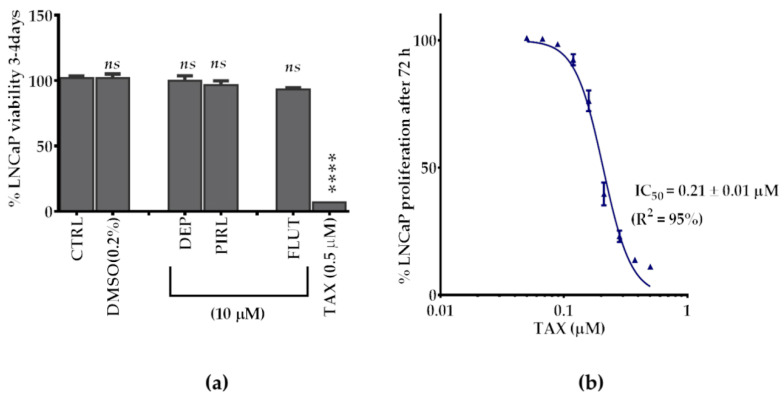
Effects of the clinical standards on the LNCaP cells. (**a**) The standards included anti-PCa drugs of paclitaxel (TAX) and flutamide (FLUT), and the antidepressant drugs of PIRL (MAO-AI), and DEP (MAO-BI), which were compared to the untreated cells and the used solvent DMSO. (**b**) Dose-response and potency of the only effective standard (TAX). All data points were presented as the mean ± SEM, *n* = 4. The IC_50_ ± SEM value was averaged from the two experiments. Significance of difference between the control and treatments were determined using one-way ANOVA, followed by Dunnett’s multiple comparisons test. *ns*—non-significant, **** *p* < 0.0001.

**Figure 6 molecules-25-02257-f006:**
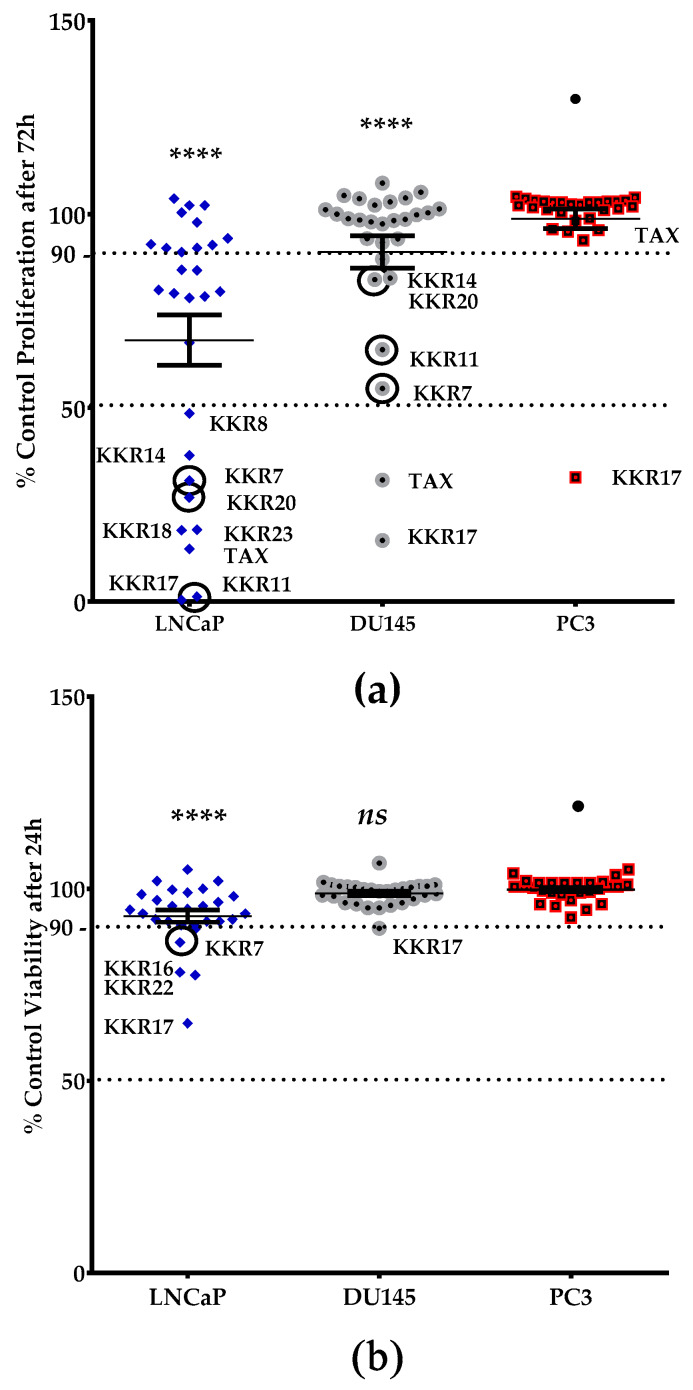
Screening the 20 KKR compounds for their cytotoxic and antiproliferative effects on THE three prostate cancer cell lines (PCa). Each 50 μM KKR-treated group of LNCaP, DU145, and PC3 cells was compared to the control of THE untreated cells. Cell viability was measured after 24 h (**a**), while cell growth was measured after 72 h (**b**) of incubation. TAX of 0.25 μM was used as a standard antiproliferative drug. Circled data points represent the repeatedly potent KKR compounds with anticancer effect, in addition to their MAO-A inhibitory activity. For all cell lines examined, every single data point represents the mean of percentage control total resazurin reduction by the viable cells (*n* ≥ 2, SEM ≤ 20%). The significance of the difference between the least affected PC3 group (•) and other cells was determined using two-way ANOVA, followed by Dunnett’s multiple comparisons test; *ns*—non-significant, **** *p* < 0.0001.

**Figure 7 molecules-25-02257-f007:**
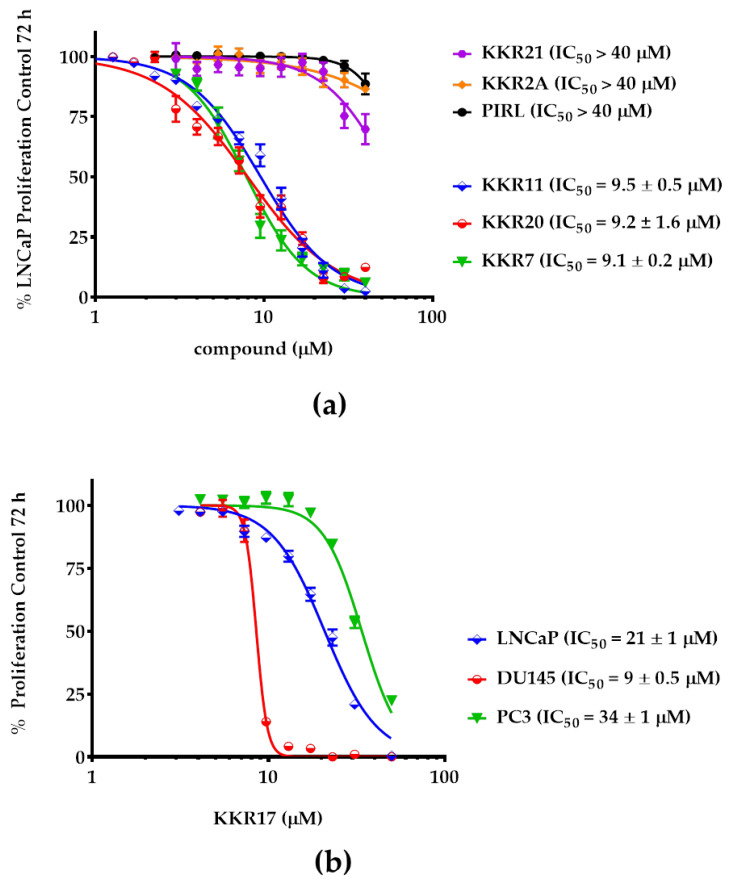
Proliferation inhibitory potencies of the top active KKRs against PCa cells. (**a**) Top five KKRs with MAO-A inhibitory activities that are diverse in their LNCaP cell inhibition. Illustrated as a dose-response comparison of KKRs with the clinical standard PIRL, KKR11, 20, and 7, were likewise potent (*p* > 0.05) while KKR21, 2A, and PIRL were not effective. (**b**) KKR17, although inactive against MAOs, showed a different pattern of behavior by inhibiting all tested PCa cell lines and affecting DU145 the most. All data points were presented as the mean ± SEM, *n* ≥ 4. IC_50_ ± SEM values were averaged from the two experiments. The significance of difference between the IC_50_s of the three active KKRs was determined using one-way ANOVA, followed by Dunnett’s multiple comparisons test.

**Figure 8 molecules-25-02257-f008:**
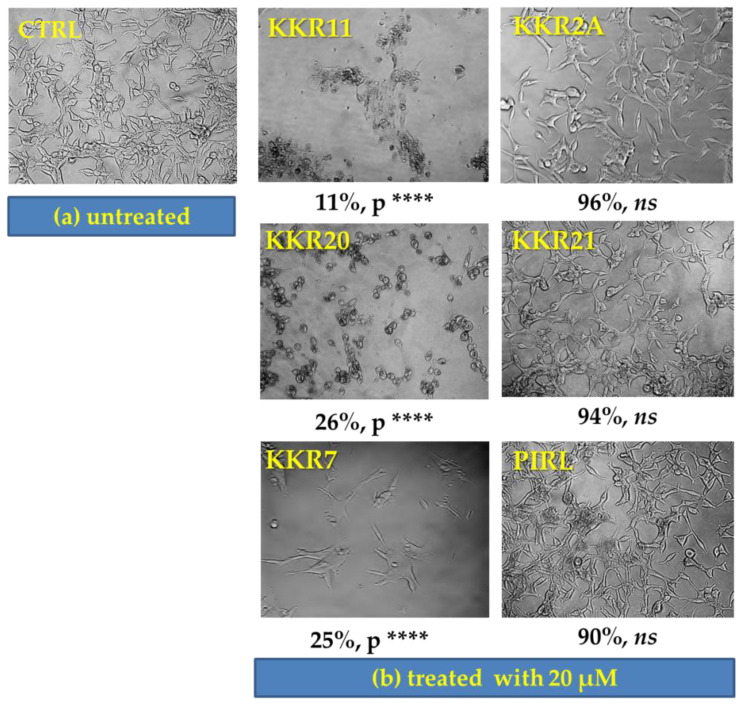
Morphological alterations of the growing LNCaP cells after five days of treatment with KKR compounds and standard PIRL; images of the control (**a**) and treated cells, (**b**) were captured using phase contrast 20X. Percentage of the treated groups was compared to the viable control (*n* ≥ 3) using one-way ANOVA followed by Dunnett’s multiple comparisons test. *ns*—non-significant, **** *p* < 0.0001.

**Figure 9 molecules-25-02257-f009:**
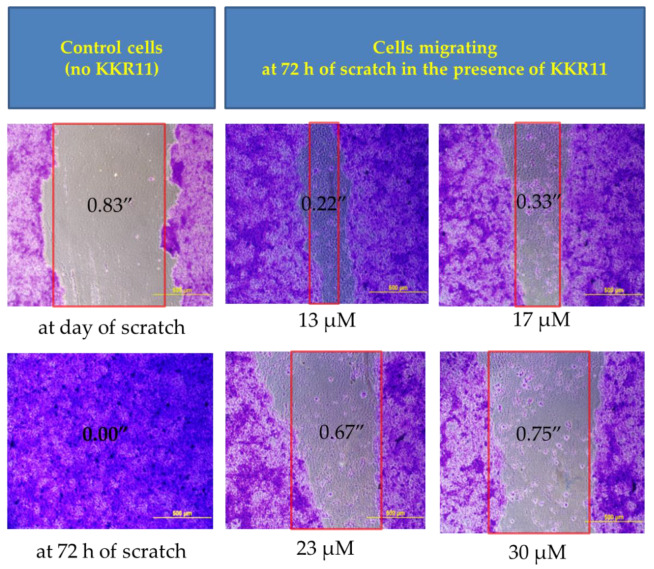
Migration of metastatic PCa cells in the presence of KKR11. The treated cells lost their ability to migrate to the space gap. DU145 cells of similar densities and incubation time were subject to scratch assay for mimicking cell migration. Scratches of original widths were fixed at the beginning of the assay. Treated cells were incubated for 72 h until the counterpart controls gaps are closed (*n* = 3). All tested cells were fixed, stained, and imaged (phase contrast, 10X). The scratch widths were analyzed. The red lines represent the approximate cell front.

**Table 1 molecules-25-02257-t001:** A series of twenty KKR flavonoid derivatives synthetic structures and their IUPAC names; eleven flavones: KKR1, KKR3, KKR4, KKR5, KKR7, KKR8, KKR9, KKR10, KKR19, KKR21, and KKR22; two chromones: KKR16 and KKR2A; and seven chalcones: KKR11, KKR12, KKR14, KKR17, KKR18, KKR20, and KKR23.

Code	Structure	IUPAC Name
KKR-1	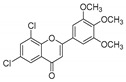	6,8-dichloro-2-(3,4,5-trimethoxyphenyl)-4*H*-chromen-4-one
KKR-2A	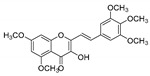	(*E*)-3-hydroxy-5,7-dimethoxy-2-(3,4,5-trimethoxystyryl)-4*H*-chromen-4-one
KKR-3	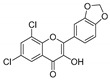	2-(benzo[*d*][1,3] dioxol-5-yl)-6,8-dichloro-3-hydroxy-4*H*-chromen-4-one
KKR-4	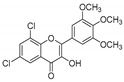	6,8-dichloro-3-hydroxy-2-(3,4,5-trimethoxyphenyl)-4*H*-chromen-4-one
KKR-5	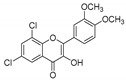	6,8-dichloro-2-(3,4-dimethoxyphenyl)-3-hydroxy-4*H*-chromen-4-one
KKR-7	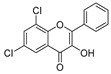	6,8-dichloro-3-hydroxy-2-phenyl-4*H*-chromen-4-one
KKR-8	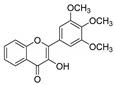	3-hydroxy-2-(3,4,5-trimethoxyphenyl)-4*H*-chromen-4-one
KKR-9	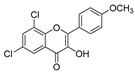	6,8-dichloro-3-hydroxy-2-(4-methoxyphenyl)-4*H*-chromen-4-one
KKR-10	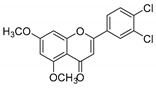	2-(3,4-dichlorophenyl)-5,7-dimethoxy-4*H*-chromen-4-one
KKR-11	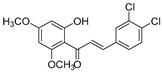	(*E*)-3-(3,4-dichlorophenyl)-1-(2-hydroxy-4,6-dimethoxyphenyl)prop-2-en-1-one
KKR-12	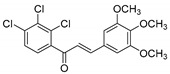	(*E*)-1-(2,3,4-trichlorophenyl)-3-(3,4,5-trimethoxyphenyl)prop-2-en-1-one
KKR-14	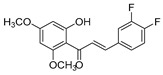	(*E*)-3-(3,4-difluorophenyl)-1-(2-hydroxy-4,6-dimethoxyphenyl)prop-2-en-1-one
KKR-16	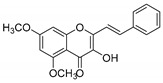	(*E*)-3-hydroxy-5,7-dimethoxy-2-styryl-4*H*-chromen-4-one
KKR-17	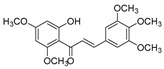	(*E*)-1-(2-hydroxy-4,6-dimethoxyphenyl)-3-(3,4,5-trimethoxyphenyl)prop-2-en-1-one
KKR-18	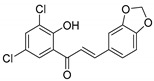	(*E*)-3-(benzo[*d*][1,3]dioxol-5-yl)-1-(3,5-dichloro-2-hydroxyphenyl)prop-2-en-1-one
KKR-19	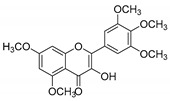	3-hydroxy-5,7-dimethoxy-2-(3,4,5-trimethoxyphenyl)-4*H*-chromen-4-one
KKR-20	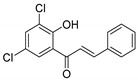	(*E*)-1-(3,5-dichloro-2-hydroxyphenyl)-3-phenylprop-2-en-1-one
KKR-21	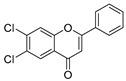	6,7-dichloro-2-phenyl-4*H*-chromen-4-one
KKR-22	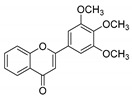	2-(3,4,5-trimethoxyphenyl)-4*H*-chromen-4-one
KKR-23	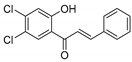	(*E*)-1-(4,5-dichloro-2-hydroxyphenyl)-3-phenylprop-2-en-1-one

**Table 2 molecules-25-02257-t002:** The effects on enzyme maximum velocity (V_max_) and the Michaelis constant (K_m_) average values with increasing KKR concentration: X-intercepts of the KKR linear regressions represent the K_m_ value changes; and the Y-intercepts of the KKR linear regressions (minimum and maximum average means shown) represent the V_max_ value changes.

Inhibitor	KKR21			KKR11		
Parameter	Average (μM)	%R^2^	*p* Value	Average (μM)	%R^2^	*p* Value
K_m_ (μM)						
0	**3.85 ± 0.44**	99		**0.98 ± 0.05**	93	
IC_50_/4	5.88 ± 0.35	99	******	1.59 ± 0.10	95	******
IC_50_	7.69 ± 0.59	99	*******	1.64 ± 0.03	95	*******
IC_50_ × 4	14.29 ± 2.04	99	********	2.44 ± 0.24	97	*******
V_max_(U/mL/h)	1.38 × 10^−6^–1.43 × 10^−6^		**ns**	3 × 10^−6^–4 × 10^−6^		**ns**
Alpha	**>>1**			**>>1**		
Ki (μM)	**0.0017**			**0.0074**		

Alpha parameter validated competitiveness. The inhibitor-constant (K_i_) was calculated using the competitive model, K_i_ = IC_50_/(1 + [S]/K_m_). Data points were expressed as the mean ± SEM, representing more than two experiments with *n* = 3 each. The significance of the difference between the control (in bold) and treatments was determined using a one-way ANOVA, followed by Dunnett’s multiple comparisons test, and the two data sets were compared using two-way ANOVA followed by Sidak’s multiple comparisons test. ns—non-significant (*p*-value for V_max_ was 0.99, and 0.42 for KKR21 and KKR11, respectively); ** *p* <0.01, *** *p* < 0.001, and **** *p* < 0.0001.

**Table 3 molecules-25-02257-t003:** Predicted responsible reversible interactions between MAO-A active site residues and KKR21 and KKR11; involved amino acid residues, cofactor, and water molecules in KKR21, KKR11, and the standard ligand harmine (2Z5X). The hydrophobic interactions are compared to 2Z5X, which was re-docked within the same site.

Non-Covalent Interaction	Shared in KKR21, KKR11 and 2Z5X	KKR21 and KKR11	with KKR11	with 2Z5X
Amino acid residue	ILE180A, ILE325A, ILE335A	ASN181A	ALA111A	THR336A
	LEU337A, PHE208A, PHE352A		LEU97A	
	TYR69A, TYR407A, TYR444A *			
	GLN215A, CYS323A, MET350A			
other molecules	HOH726, HOH739, HOH746	FAD 600A		
	HOH766, HOH805	HOH710		

* This residue shared a hydrogen-bond with KKR11. Key: LEU—leucine, ILE—isoleucine, TYR—tyrosine, PHE—phenylalanine, GLN—glycine, CYS—cysteine, MET—methionine, ASN—asparagine, ALA—alanine, and THR—threonine.

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
