# Peer review of "The Antiproliferative Effects of Flavonoid MAO Inhibitors on Prostate Cancer Cells"

_molecules, 2020, doi:10.3390/molecules25092257_

Round 1
Reviewer 1 Report
This study intends to targeting depression in PCa. These derivatives do exhibit MAO inhibitory effects and anticancer effects, while the work/results are not well presented as the authors seemed to be eager to show all the results, even those that were not closely related to the topic. Please reorganize this study, and present it topic focused.
1, The statement of "we synthesized twenty flavonoid derivative compounds (KKRs)" in the Abstract is inaccurate.
2, Please re-write the Abstract to draw clear rationale/connections between MAO (and its inhibitors) with PCa.
3, "allowing for high mortality and morbidity rates calling for more efficient and less controversial treatments." what this means?
4, Language needs to be moderate changed.
5, Unless a model of PCa with depression is constructed and tested upon treatments of KKRs, to me, the scientific significance is low. Although flavonoids can be used to treat both depression and at the same time cancer (in lab study only perhaps), while they are too weak to be considered as the drug candidates.
6, It's my opinion that SAR (including docking study) could be excluded while I will reconsider it if the authors can revise it in a rational way.
7, Structures of KKR21-23 are missing.
8, Please provide the rationale of the dose chosen in Section 2.3.4.
9, " Our objective is to find a safe RIMA compound", where are the data related "safe"?
Author Response
Comments and Suggestions for Authors with point by point responses
Dear Editor: We appreciate the constructive criticisms, questions, and comments of the reviewers, and we have addressed each of their concerns as outlined below.
Responses to Reviewer # 1
Comment 1: This study intends to be targeting depression in PCa. These derivatives do exhibit MAO inhibitory effects and anticancer effects, while the work/results are not well presented as the authors seemed to be eager to show all the results, even those that were not closely related to the topic. Please reorganize this study and present it topic focused.
Reply: This study intends to mainly target PCa with depression, not mainly depression as in the discussion
Line 371-472 “for managing PCa with depression,”.
We clarified this in the introduction Line
From Line 117-118 “In the current study, we are identifying synthetic MAO-AIs for the possible therapeutic use in the management of PCa and depression”
To Line 133-134 “In the current study, we are identifying synthetic MAO-AIs for the possible therapeutic use in the management of PCa with depression”
The study is to evaluate the hypothesis that all MAO-Is can have anti-PCa effects (clarified in the next paragraph). This requires the topic of in vitro investigation to be on two unseparated main sections; PCa models and MAO isozymes inhibitory effects models. Although the topic becomes large, the investigations collectively converge to answer the main scientific inquiry related questions. Thus, we think that these results relate to each other and are strongly associated to serve the point of this work, without which, conclusions will have no evidence.
However, to make the presentation focus clearer, the authors modified the following paragraph in the introduction (can be tracked in the revised manuscript):
From (original manuscript; line 117) “In the current study, we are identifying synthetic MAO-AIS for the possible therapeutic use in the management of PCa and depression, without the side effects of not metabolizing other essential and dietary monoamines. In our search for compounds with these multifaceted actions, we investigated 20 synthesized compounds (KKRs) (Fig.1) against recombinant human MAO isozymes (hMAO-A and hMAO-B) activities and evaluated their antiproliferative activities against three human PCa metastatic cell lines. Our objective is to find a safe RIMA compound with antiproliferative and anti-metastatic effects against aggressive PCa cells.”
To (revised manuscript Line 133) “In the current study, we are identifying synthetic MAO-AIs for the possible therapeutic use in the management of PCa and depression. Also, to predict the possible side effects of having consequential excessive essential and dietary monoamines that MAO-AIs may cause, we are determining their safety in PCa treatment by investigating the top MAO-AIs mode of inhibition. In our search for compounds with these multifaceted actions, we investigated 20 synthesized compounds (KKRs) (Table.1) against recombinant human MAO isozymes (hMAO-A and hMAO-B) activities and evaluated their antiproliferative effects against three human PCa metastatic cell lines. Our main objective is to find a reversible—and thus—safe RIMA compound with antiproliferative and anti-metastatic effects against aggressive PCa cells, with an endeavor to understand the underlying molecular mode of inhibitions. With multiple various in vitro screens conducted, we evaluate whether all MAO-Is can have anti-PCa effects.”
Comment 2: The statement of "we synthesized twenty flavonoid derivative compounds (KKRs)" in the Abstract is inaccurate.
Reply: The synthesis of the twenty KKRs was in our laboratories. However, we changed the expression for correction as follows:
Revised Line 25 (in the revised manuscript) “...we synthesized twenty flavonoid derivatives (KKRs)…”
Comment 3: Please re-write the Abstract to draw clear rationale/connections between MAO (and its inhibitors) with PCa.
Reply: Thank you for the comment. The authors made the following changes to draw these connections as follows (can be tracked in the revised manuscript):
Revised manuscript lines 20-23: “Abstract: Prostate cancer (PCa) patients commonly experience clinical depression. Recent reports indicated that monoamine oxidase-A (MAO-A) levels elevate in PCa, and antidepressant MAO-Is show anti-PCa properties. In this work, we aim to find potential drugs for PCa patients suffering from depression by establishing novel anti-PCa reversible monoamine oxidase-A inhibitors (MAO-AIs / RIMA)”
Comment 4: "allowing for high mortality and morbidity rates calling for more efficient and less controversial treatments." what this means?
Reply: Thank you for the comment. The statement and paragraph have been clarified as follows (can be tracked in the revised manuscript):
Revised Lines 69-73: “benefits are yet clinically limited by facing increasing cancer resistance that allows for high mortality and morbidity rates. These limitations raised calls for the demand for more research on more efficient and less controversial possible treatments [13]. Indeed, with ADT intolerable limitations, better regimens and multifunctional treatments against PCa becomes a necessity”
Comment 5: Language needs to be moderate changed.
Reply: The authors made language and scientific reviews to improve the language and enhance clarity. Authors uses a suitable software for language review. All changes were made in the revised manuscript and highlighted in yellow. (all changes are tracked in the manuscript)
Comment 6, Unless a model of PCa with depression is constructed and tested upon treatments of KKRs, to me, the scientific significance is low. Although flavonoids can be used to treat both depression and at the same time cancer (in lab study only perhaps), while they are too weak to be considered as the drug candidates.
Reply: The authors totally agree that KKRs should be tested in vivo on PCa model with depression to indicate their clinical significance on PCa patients who suffer depression. However, our in vitro screen is a comprehensive step earlier; here, we constructed the evidence bases that are essential for indicating the significance of future in vivo and clinical studies. Because these KKRs are synthetic and their specific activities cannot be predicted from nutrition data, this work provides the rationale behind predicting and understanding of future KKRs clinical investigations. Thus, the possible scientific and therapeutic significance of understanding different KKRs activities and mechanisms against PCa and MAO will not be known without this investigation. Moreover, KKRs activities were compared to clinical standards currently used in prostate cancer (paclitaxel) and in clinical depression (pirlindole and deprenyl). Our results indicated better and comparable outcome from KKRs. Thus, we presume that top highly potent KKRs will reflect high scientific significance in future studies.
The scientific significance for different flavonoid structures to be considered as drug candidates is under ongoing investigations that are expected to continue (Kikuchi, Yuan, Hu, & Okazaki, 2019). Like any compound a potential drug faces challenges when proposed for potential clinical trial, so are the varied flavonoid compounds. Although many flavonoid structures have clinical bioavailability challenges due to solubility issues, it is well common to have solubility issues in drugs. For that, many new manipulative pharmaceutical techniques were and are proposed (Kikuchi, Yuan, Hu, & Okazaki, 2019). Also, the reversibility investigation in our work is preformed to reveal drug-drug or food-drug interactions that are one of the challenges of flavonoids, especially, the MAO-Is. The flavonoid structure and literature review effects were very inspiring to accomplish our aim as was discussed in the introduction (Lines 107-117) and throughout the discussion section. Indeed, the inhibitory potency in many reported flavonoid-MAO-Is showed reversible and potent MAO inhibitions within the nanomolar range, which satisfy our main strategic goals for achieving new future PCa alternative therapies.
The authors added the following statements in the revised manuscript that indicate the progress and findings of clinical trials despite the challenges (changes can be tracked in the revised manuscript):
Revised Lines 119-121: “Previous studies support that flavonoids with multidrug resistance (MDR) inhibitory properties in cancer embody a promising strategy for future chemopreventive therapy [30]”
Revised Lines 127-131: “Clinically, despite the challenges to reflect the in vitro achievements, research is on the go for flavonoids such as quercetin, genistein, and epigallocatechin gallate for their anticancer effects. Indeed, flavonoids nutraceuticals, combined with common chemotherapeutics for chemopreventive anticancer activities, recently showed progressively promising results (Kikuchi, Yuan et al. 2019).”
Also, the significance of this work is not limited to its choice of structure; this work could be the first of its kind to establish potential drug candidate for depressed PCa patients, by combining their model in one study, as most screening investigations focus on either anticancer effect or MAO inhibition, but not combined and compared in one study. The closest two reports to our work we found have suggested the potential use of MAO-Is for many diseases including depression and PCa. But they did not test PCa models (revised Line 545-547; ref. 57 and 58 in the revised manuscript). That makes our work unique and promising. We indicated this understanding throughout the discussion when pointing out synthetic and natural flavonoids related reports as follows in the highlights
Original Lines 488-497:
“Other synthesized derivatives of 2-imidazoline, pyrrolo[3,4-f]indole-5,7-dione, and indole-5,6-dicarbonitrile were reported as MAO-Is and were suggested for the treatment of different diseases including depression or PCa [50,51]. In a comparison of KKR11 to other flavonoid structures, specific natural flavonoids have shared multiple actions related to PCa or depression in a variety of studies. An example is genistein, a soybean isoflavone, which proved a reversible MAO-A inhibition [39], improved cognitive functions in males [52], and exhibited antioxidant and anti-cancer effects [53,54]. In PCa cells, genistein protected from the aging-related DNA damage [55] and inhibited epithelial-to-mesenchymal transition [56] and LNCaP invasiveness [57]. Consequently, KKR11 promises a beneficial treatment for metastatic aggressive PCa that is concurrent with depression. MAO-AIs potential to eliminate metastasis grants them to be novel therapeutics for PCa and depression patients, and that needs to be emphasized.”
Finally, the authors proposed the active KKRs as potential drug candidates but not drug candidates, as it is too early to make this conclusion. This was already mentioned throughout the original manuscript in the following highlighted positions:
Original Lines 37-38: “The obtained results indicated that the flavonoid derivative KKR11 could present a novel candidate for PCa patients with depression” Here we used could indicate it is too early to say a novel candidate.
Original Lines 345-346: “This behavior highlights the compound as a great candidate for further studies”
Lines 454-455: “KKR structures embrace multifunctional candidate drugs for managing PCa with depression,”
Original Lines 754-755: “and thus, proposed KKR17 as a candidate for further investigations on PCa and its mechanism.” Here, PCa means continuing in vitro future studies
Original Lines 765-767: “None the less, KKR21 selectivity and high potency is comparable to the clinical RIMA drug PIRL, making it a potential antidepressant drug candidate with fewer side effects but not for PCa treatment.”
The authors changed the following for clarification and consistency:
Original Lines 436-437: (can be tracked in the revised manuscript)
From “making it an exceptional candidate for PCa patients with depression.”
To revised lines 445 “making it an exceptional potential candidate for PCa patients with depression.”
Comment 7, It is my opinion that SAR (including docking study) could be excluded while I will reconsider it if the authors can revise it in a rational way.
Reply: One of the objectives of this study is to understand the molecular mechanisms of differently active KKRs that showed different or similar activities against our PCa models and MAO-A inhibition and mode of action. To answer what active group or interaction contributes to a specific active KKR behaviour, we made our scientific predictions through docking and SAR studies. The rational for use of docking studies and SAR in this work is already mentioned in the results section in the following lines:
Original Lines 235-237 “Docking studies were conducted to rationalize the molecular mechanisms of the reversible mode of inhibition of KKR21 and KKR11. The responsible interactions of the molecules with the crystal structure of the human MAO-A monopartite active site were predicted (Fig. 4).”
For better presentation of the work, authors revised the manuscript to strengthen the rational of SAR and docking studies by adding/modifying the following highlighted lines (can be tracked in the reviewed manuscript):
In abstract.
Revised Lines 22-24: “In this work, we aim to find potential drugs for PCa patients suffering from depression by establishing novel anti-PCa reversible monoamine oxidase-A inhibitors (MAO-AIs / RIMA); with an endeavor to understand their mechanism of action.”
Revise Lines 27-29: “MAO-A-kinetics, molecular docking, SAR, cell morphology, and cell migration were investigated for the most potent compounds.”
Revised Lines 33-44: “Furthermore, KKR21 and KKR11 inhibited MAO-A competitively (Kis ≤ 7.4 nM). Molecular docking of the two compounds predicted shared hydrophobic and distinctive hydrophilic interactions—between KKR molecule and MAO-A amino acid residues—to be responsible for their reversibility. The combined results and SAR observations indicate that the presence of specific active groups—such as chlorine and hydroxyl groups—are essential in certain MAO-AIs with anti-PCa effects. Also, MAO-A inhibition is more associated to the anti-PCa property than MAO-B. Distinctively, KKR11 [(E)-3-(3,4-dichlorophenyl)-1-(2-hydroxy-4,6-dimethoxyphenyl) prop-2-en-1-one], exhibited anti-metastatic effects on DU145 cell line. The chlorine substitution groups may play vital roles in KKR11 multiple actions. The obtained results indicated that the flavonoid derivative KKR11 could present a novel candidate for PCa patients with depression through safe non-selective potent inhibition of MAOs.”
In the introduction.
Revised Lines 136-143: “In our search for compounds with these multifaceted actions, we investigated 20 synthesized compounds (KKRs) (Table.1) against recombinant human MAO isozymes (hMAO-A and hMAO-B) activities and evaluated their antiproliferative effects against three human PCa metastatic cell lines. Our main objective is to find a reversible—and thus—safe RIMA compound with antiproliferative and anti-metastatic effects against aggressive PCa cells, with an endeavor to understand the underlying molecular mode of inhibitions. With multiple various in vitro screens conducted, we evaluate whether all MAO-Is can have anti-PCa effects.”
In results.
Revised Lines 160-162: “Comparing SAR of the top active five structures to SAR of their less or non-active KKR analogues, we identified some specific functional groups in each active KKR that are crucial for causing or increasing the MAO inhibitory effects.”
In conclusion.
Revised Lines 781-783: “However, the results and SAR studies indicate that only certain MAO-AIs can have anti-PCa effects, and they do not have to be selective to MAO-A to exhibit the anti-PCa property.”
Revised Lines 803-806: “The mechanism is possibly by the non-selective potent and reversible MAO-A inhibitory property that reduces oxidative stress and impedes PCa malignancy promotion and aggressiveness. The KKR11 shape, lipophilicity, and the chlorine groups may play a significant role in these in vitro actions.”
Comment 8, Structures of KKR21-23 are missing.
Reply: The structures are presented with their IUPAC names listed in an ascending numerical order in Table 1. (Original manuscript Line 144-147). The names are also mentioned in the table caption.
Comment 9: Please provide the rationale of the dose chosen in Section 2.3.4.
Reply: In our manuscript, section 2.3.4. of Results is shown below:
“2.3.4. Three MAO-A inhibiting KKRs affected LNCaP cells morphology differently
The LNCaP cell morphological alterations, along with their proliferative growth, were monitored for several days of exposure to the five selected KKRs and PIRL. The morphological effects of the selected KKRs, at their 20 μM concentration level, were highly pronounced after five-days of exposure (Fig. 9).”
The rational of the 20 μM KKR was explained in the Materials and methods in the following section, where it is highlighted:
“4.3.3.2. Imaging the proliferative LNCaP after five-days
The morphological alterations of LNCaP cells have been imaged, and their percentage proliferation rate was measured after five-days of incubation. The method of subculture and treatment were similar to our antiproliferation assay with slight modifications. Briefly, cells were seeded in transparent flat-bottom 96-well microplates with 10,000 cells/175 μL/well of RPMI1460 experimental media. Following overnight incubation, 20 μM concentration level (~ 2 x IC50) for each of KKR11, 20, 7, 2A, 21, and the standard PIRL were ...”
However, to increase clarification, the authors made changes on section 2.3.4. as follows; (can be tracked in the revised manuscript):
Revised Lines 389-394:
“2.3.4. Three MAO-A inhibiting KKRs affected LNCaP cells morphology differently
The LNCaP cell morphological alterations, along with their proliferative growth, were monitored for several days of exposure to the five selected KKRs and PIRL. Fixed 20 μM concentrations were chosen to guarantee higher concentration than any IC50 of any tested compound and to ease the comparison between compound effects. The morphological effects of the selected KKRs, were highly pronounced after five-days of exposure (Fig. 9).”
Comment 10 "Our objective is to find a safe RIMA compound", where are the data related "safe"?
Reply: The data related to “safe” are the mode of inhibition (kinetics data) and the docking studies to rationale it. The safety issue of MAO inhibition is mentioned in the introduction. The authors made the modifications for clarification in the introduction paragraph, as previously shown (original line 122) to be as follows (can be tracked in the revised manuscript).
Revised Line 140:
“Our main objective is to find a reversible—and thus—safe RIMA compound”
References
Kikuchi, H., Yuan, B., Hu, X., & Okazaki, M. (2019). Chemopreventive and anticancer activity of flavonoids and its possibility for clinical use by combining with conventional chemotherapeutic agents. Am J Cancer Res, 9(8), 1517-1535.
Reviewer 2 Report
In the present papers Authors described the anticancer effects of newly synthetized flavonoid derivative compounds (KKRs) in three different prostate cancer cell lines and investigated KKRs ability to inhibit mono-amine oxidase isoenzymes. For the most promising compounds Authors investigated MAO-A and B inhibition, molecular docking, the cytotoxic, antiproliferative and anti-metastatic effects. Authors proposed KKRs as a complementary treatment for both PCa and depression, due to their ability to inhibit MAO that contributes to PCa development and depression. Authors finally identify a most promising compound KKR11 that shows MAO-A and -B inhibition, antiproliferative (LNCaP, DU145) and antimetastatic activity (DU145) in two prostate cancer cell lines. KKR11 emerges as an interesting candidate to treat prostate cancer and depression contemporary, but other candidates emerge as interesting MAO-A inhibitors or for their anticancer effects (KKR17). In my opinion the paper is well written and presents in vitro promising results, thus, deserves publication in Molecules after minor revision.
Introduction section provides all necessary information to follow the experimental design and to understand the cutting-edge aim of this in vitro study.
Line 262: What the Authors mean with “KKRs affected PCa cells”? Please specify, i.e. KKRs cytotoxic and antiproliferative effects in PCa cells. The same in paragraph 2.3.1. I understand that Authors would prefer to talk about effects in general (due to the anticancer drugs and antidepressant drugs used as standard), but to help the reader it is possible to specify antiproliferative effects because is the final biological endpoint they observed.
Line 287: What Authors means with “The nontoxic clinical drug TAX”? Please specify.
Line 292: please correct punctuation
It would be nice to speculate in the discussion section about the difference in term of antiproliferative effects of KKRs compounds in the three different cell lines. Why PC3 are the less affected by the antiproliferative effects of KKR17? Only the most promising derivative without MAO inhibitory activity has showed antiproliferative effect. Please speculate.
Author Response
Comments and Suggestions for Authors with point by point responses
Dear Editor: We appreciate the constructive criticisms, questions, and comments of the reviewers, and we have addressed each of their concerns as outlined below.
Responses to Reviewer #2
In the present paper, the Authors described the anticancer effects of newly synthetized flavonoid derivative compounds (KKRs) in three different prostate cancer cell lines. They investigated the KKR's ability to inhibit monoamine oxidase isoenzymes. For the most promising compounds Authors investigated MAO-A and B inhibition, molecular docking, the cytotoxic, antiproliferative, and antimetastatic effects. Authors proposed KKRs as a complementary treatment for both PCa and depression, due to their ability to inhibit MAO that contributes to PCa development and depression. Authors finally identify the most promising compound KKR11 that shows MAO-A and -B inhibition, antiproliferative (LNCaP, DU145), and antimetastatic activity (DU145) in two prostate cancer cell lines. KKR11 emerges as an interesting candidate to treat prostate cancer and depression contemporary, but other candidates emerge as interesting MAO-A inhibitors or for their anticancer effects (KKR17). In my opinion, the paper is well written and presents in vitro promising results; thus, it deserves publication in Molecules after minor revision.
Comment 1:
The introduction section provides all the necessary information to follow the experimental design and to understand the cutting-edge aim of this in vitro study.
Reply:
Thank you for your comment. Slight modifications highlighted in yellow were made to improve clarity and language in the introduction and discussion (changes are tracked in the revised manuscript)
Comment 2
- Line 262: What the Authors mean with
- “KKRs affected PCa cells”? Please specify, i.e., KKRs cytotoxic and antiproliferative effects in PCa cells. The same in paragraph 2.3.1. I understand that Authors would prefer to talk about effects in general (due to the anticancer drugs and antidepressant drugs used as standard), but to help the reader, it is possible to specify antiproliferative effects because it is the final biological endpoint they observed.
Reply:
The authors agree with the reviewer's advice. Please see the different changes we did in the manuscript as follows (changes are tracked in the revised manuscript)
Revised manuscript Lines 285-286:
“2.3. KKRs cytotoxic and antiproliferative effects in PCa cells
2.3.1. Standards effects on PCa cells proliferation.”
Comment 3
- Line 287: What Authors means with “The nontoxic clinical drug TAX”? Please specify.
Reply:
Nontoxic means it did not affect the viability of any cell line tested in our cytotoxicity assays. The nontoxic effect of TAX was repeatedly observed on our PCa screens data. The expression “nontoxic” mentioned twice in the same section of the results:
section 2.3.2, original manuscript Line 292: “The nontoxic clinical drug, TAX, was used as a positive antiproliferative standard.”
section 2.3.2, original Lines 336-337: “Exceptions were four KKRs that affected LNCaP cells (KKR17, KKR22, KKR16, and KKR7). The viability of the other two types of cells was not affected. The nontoxic behavior of KKRs with hMAO-A inhibitory properties was similar to TAX.”
To mitigate confusion, the authors deleted the “nontoxic” that mentioned first to become as follows:
Revised Line 310: “The clinical drug, TAX, was used as a positive antiproliferative standard.”
Comment 4
Line 292: please correct punctuation
Reply:
Original Line 292 “The nontoxic clinical drug TAX was used as a positive antiproliferative standard.”
Was corrected to
Revised Line 310: “The clinical drug, TAX, was used as a positive antiproliferative standard.”
Comment 5
It would be nice to speculate in the discussion section about the difference in terms of antiproliferative effects of KKRs compounds in the three different cell lines. Why are PC3 less affected by the antiproliferative effects of KKR17? Only the most promising derivative without MAO inhibitory activity has shown an antiproliferative effect. Please speculate.
Reply:
It is interesting that MAO-Is KKRs did not have any detectable effects on PC3, while the only compound that affected the PC3 cells, chalcone KKR17, was not an MAO-I, and yet, PC3 continued to be the most resistant. The speculations for PC3 resistance to KKRs were added to the Discussion section in three consecutive paragraphs that also connect the PC3 resistance to LNCaP vulnerability to KKRs and DU145 vulnerability to KKR17. The added paragraphs are below (please note that reference numbers have changed accordingly in the revised manuscript and changes can be tracked):
Revised Lines 592-520:
“It is interesting that MAO-Is KKRs did not have any detectable effects on PC3, while the only compound that affected the PC3 cells, chalcone KKR17, was not an MAO-I, and yet, PC3 continued to be the most resistant. All of PCa cells under investigation are epithelial cell lines that express MAO-A and not MAO-B [1]. By looking at the MAO-A levels in these cell lines we find that the cellular phenotype LNCaP expresses both MAO-A transcript and protein in much greater levels than that of DU145 and PC3 cells [1]. That may explain LNCaP susceptibility to a higher number and high potency of KKRs compared to DU145 and PC3 cells. Conversely, PC3 cells very low MAO expressions may increase resistance to KKRs in this aspect. Nevertheless, the MAO-independent antiapoptotic effects of KKR17, once again, approve a different mechanism of affecting the three cell lines regardless of their MAO expression or activity.
With their low MAO-A expression, PC3 cells may possess other characters that increase their resistance to KKRs in general. PC3 cells exhibit the characteristics of the highly malignant but rare prostatic small cell carcinoma, which differs from LNCaP that represents the common prostatic adenocarcinoma characters [2]. Indeed, there are characters of PC3 cells that increase its resistance compared to LNCaP cells, such as the positivity for the stem cell-associated marker CD44, which enables PC3 cells for better intercellular interactions and migration [2]. More importantly, only PC3 cells have low levels of topoisomerase 2α (TOP2α), the double-strand breaker that has been associated with chemotherapy resistance [3]. While KKR17 has demonstrated great potentials as a cytotoxic antiproliferative agent in all of the three PCa phenotypes but with the least effects on PC3, KKR17 highest potency was against DU145 cells.
Conversely to PC3, the fast proliferative DU145 cells express the highest levels of TOP2α, which is considered a target for main anticancer drugs [4] and other versatile catalytic TOP2α inhibitors [5]. Indeed, KKR17 dose-dependent inhibition of DU145 cell proliferation displayed a sharp slope with an angle that varied from LNCaP and PC3 slopes of inhibition (Fig.7b) which hypothetically reflects its proportion to TOP2α. Regardless, several flavonoids previously reported affective against DU145, such as genistein, apigenin, chrysin, and quercetin [6]. As a flavonoid derivative, KKR17 may have promoted apoptosis, necrosis, cell cycle arrest, or autophagy to inhibit the cells proliferative growth, particularly, the androgen-independent DU145 phenotype, and that should be further investigated.”
References:
- Gordon, R.R.; Wu, M.; Huang, C.Y.; Harris, W.P.; Sim, H.G.; Lucas, J.M.; Coleman, I.; Higano, C.S.; Gulati, R.; True, L.D., et al. Chemotherapy-induced monoamine oxidase expression in prostate carcinoma functions as a cytoprotective resistance enzyme and associates with clinical outcomes. PloS one 2014, 9, e104271.
- Tai, S.; Sun, Y.; Squires, J.M.; Zhang, H.; Oh, W.K.; Liang, C.Z.; Huang, J. Pc3 is a cell line characteristic of prostatic small cell carcinoma. The Prostate 2011, 71, 1668-1679.
- Fry, A.M.; Chresta, C.M.; Davies, S.M.; Walker, M.C.; Harris, A.L.; Hartley, J.A.; Masters, J.R.; Hickson, I.D. Relationship between topoisomerase ii level and chemosensitivity in human tumor cell lines. Cancer research 1991, 51, 6592-6595.
- van Brussel, J.P.; van Steenbrugge, G.J.; Romijn, J.C.; Schroder, F.H.; Mickisch, G.H. Chemosensitivity of prostate cancer cell lines and expression of multidrug resistance-related proteins. European journal of cancer (Oxford, England : 1990) 1999, 35, 664-671.
- Larsen, A.K.; Escargueil, A.E.; Skladanowski, A. Catalytic topoisomerase ii inhibitors in cancer therapy. Pharmacol Ther 2003, 99, 167-181.
- Abotaleb, M.; Samuel, S.M.; Varghese, E.; Varghese, S.; Kubatka, P.; Liskova, A.; Busselberg, D. Flavonoids in cancer and apoptosis. Cancers 2018, 11.
Round 2
Reviewer 1 Report
Thank you for the revision. The only concern remained is that the statement " we synthesized twenty flavonoid derivatives (KKRs)" is inaccurate, as the authors only screened them but not synthesized them in the current study. Plus, the synthesis work has already published over 10 years.